# Neurexin and Neuroligin-based adhesion complexes drive axonal arborisation growth independent of synaptic activity

William D Constance[1,2], Amrita Mukherjee[1], Yvette E Fisher[3,4], Sinziana Pop[1], Eric Blanc[5], Yusuke Toyama[6,7,8], Darren W Williams[1]*

[1]Centre for Developmental Neurobiology, King's College London, London, United Kingdom; [2]King's-NUS Joint Studentship Program, King's College London, London, United Kingdom; [3]Department of Neurobiology, Stanford University, Stanford, United States; [4]Department of Neurobiology, Harvard Medical School, Boston, United States; [5]Berlin Institute of Health, Berlin, Germany; [6]Department of Biological Sciences, National University of Singapore, Singapore, Singapore; [7]Temasek Life Sciences Laboratory, Singapore, Singapore; [8]Mechanobiology Institute, National University of Singapore, Singapore, Singapore

**Abstract** Building arborisations of the right size and shape is fundamental for neural network function. Live imaging in vertebrate brains strongly suggests that nascent synapses are critical for branch growth during development. The molecular mechanisms underlying this are largely unknown. Here we present a novel system in *Drosophila* for studying the development of complex arborisations live, in vivo during metamorphosis. In growing arborisations we see branch dynamics and localisations of presynaptic proteins very similar to the 'synaptotropic growth' described in fish/frogs. These accumulations of presynaptic proteins do not appear to be presynaptic release sites and are not paired with neurotransmitter receptors. Knockdowns of either evoked or spontaneous neurotransmission do not impact arbor growth. Instead, we find that axonal branch growth is regulated by dynamic, focal localisations of Neurexin and Neuroligin. These adhesion complexes provide stability for filopodia by a 'stick-and-grow' based mechanism wholly independent of synaptic activity.
DOI: https://doi.org/10.7554/eLife.31659.001

*For correspondence:
darren.williams@kcl.ac.uk

Competing interests: The authors declare that no competing interests exist.

## Introduction

Neurons are the most structurally diverse and complex cell type we know of (*Bullock and Horridge, 1965*). Their tree-like arborisations are critical for collecting, integrating and disseminating information between different synaptic partners. During development, arborisations grow dynamically and this morphogenesis sets limits on their final location, possible synaptic partners and core electrophysiological properties (*Chagnac-Amitai et al., 1990*; *Roberts et al., 2014*). Although there are recognisable cell-type specific shapes, each individual neuron has its own unique pattern of branching and connectivity. The 'generative rules' for constructing complex arborisations are believed to be encoded by genetic algorithms that play out in different developmental contexts to produce distinct morphological types (*Cuntz et al., 2010*; *Teeter and Stevens, 2011*; *Chen and Haas, 2011*; *Hassan and Hiesinger, 2015*). Which molecules and mechanisms underlie these algorithms remains a major unanswered question within neuroscience.

From observations on the earliest phases of motoneuron dendrite growth in the spinal cord Vaughn and colleagues forwarded the *synaptotropic hypothesis* (*Vaughn et al., 1974*, *1988*). The *synaptotropic hypothesis* posits that the stability of growing axonal and dendritic branches is

controlled by the selective stabilisation of processes by nascent synapses, encouraging growth into territories rich in potential synaptic partners. Berry and colleagues, working on Purkinje cells, forwarded the same idea, calling it the *synaptogenic filopodial theory* (*Berry and Bradley, 1976*). Since then, live imaging in vertebrate brains has revealed dynamic axonal and dendritic growth (*Kaethner and Stuermer, 1992*; *Wu et al., 1999*; *Jontes et al., 2000*; *Hossain et al., 2012*) where the arrival and localisation of synaptic machineries is correlated with branch stabilisation (*Alsina et al., 2001*; *Niell et al., 2004*; *Meyer and Smith, 2006*; *Ruthazer et al., 2006*). Data on *Xenopus* tectal neuron dendritic growth revealed a role for Neurexin (Nrx) and Neuroligin (Nlg) in branch dynamics, whereby Nrx-Nlg interactions are believed to direct the genesis and maturation of opposing hemisynapses, after which neurotransmission stabilises branches (*Chen et al., 2010*). The idea that synaptic transmission stabilises nascent contacts during elaboration is supported by a number of studies (*Rajan et al., 1999*; *Sin et al., 2002*; *Ruthazer et al., 2003*; *Haas et al., 2006*; *Ruthazer et al., 2006*). However, other work has suggested that activity has little impact on large-scale features of tree growth but plays a more nuanced role in refining structural connectivity during activity dependent plasticity (*Verhage et al., 2000*; *Varoqueaux et al., 2002*; *Hua et al., 2005*; *Ben Fredj et al., 2010*).

Two key questions arise from this work: firstly, are growing branches stabilised by local 'synaptogenic' events? Secondly, are such nascent synapses being 'use-tested' by conventional neurotransmission? Here we describe a new system that we have pioneered in *Drosophila* to explore these questions. Unlike in the fly embryo where arborisations are very small and built very rapidly (<4 hr), our system takes advantage of the large axonal arborisations of the pleural muscle motoneurons (PM-Mns) that are built over an extended period, during metamorphosis. Also, unlike larval motoneuron terminals, which grow incrementally to scale with the changes in muscle size during larval life, the PM-Mn axonal arborisations grow exuberantly, by trial-and-error, very much like complex neurons found in vertebrate central nervous systems.

During development we see a consistent relationship between the distribution of presynaptic machineries and branch dynamics similar to that found in the retinal ganglion cell axons in fish and frogs. Importantly, we find that branch growth is driven by dynamic complexes formed between synaptic partners that we term 'neuritic adhesion complexes' (NACs). These NACs contain Nlg1 (postsynaptically), along with Nrx, Syd1 and Liprin-α (presynaptically), and act locally to stabilise filopodia by a '*stick and grow*' mechanism without the need of synaptic vesicle release machinery or functional synapses.

## Results

### Establishing a new model for exploring complex arborisation growth

In a search for a system to study complex arbor growth live, in vivo, we identified the axonal arborisations of the motoneurons that innervate the abdominal pleural muscles of the adult fly. We refer to these as the pleural muscle motoneurons (PM-Mns), the axonal arborisations of which are large, complex and easily accessible throughout metamorphosis. Importantly, the PM-Mn system allows one to genetically manipulate either synaptic partner whilst independently imaging the other.

Each adult abdominal hemisegment (from A1-A7) contains a pair of motoneurons that exit the nerve onto the lateral body wall (*Figure 1A*). In the adult, each motoneuron forms an arborisation that innervates between 15 and 18 parallel muscle fibres that span dorso-ventrally, from the tergites to sternites (*Figure 1B*). In contrast to the rigid, target-specificity found in the larval neuromuscular system, the innervation of the pleural muscles shows variation between segments and between animals. Although no one-to-one neuron/muscle targeting is found, arborisations achieve a consistent spatial organisation with regularly sized, non-overlapping projection fields.

To gain insight into the global development of the PM-Mn axon terminals, we imaged hemi-segment A3 from 24 hr to 84 hr after puparium formation (APF) (*Figure 1C*). By 24 hr APF the majority of larval muscles, and the neurons that innervate them, are removed by programmed cell death and phagocytosis (*Currie and Bate, 1991*). The dorsally projecting peripheral nerve maintains four contacts with the epidermis during pupariation and allows the continued innervation of the persistent larval muscles. At ~24 hr APF axonal regrowth commences with the emergence of filopodia-rich growth cones from the nerve. By 34 hr APF primary branches are established and festooned with

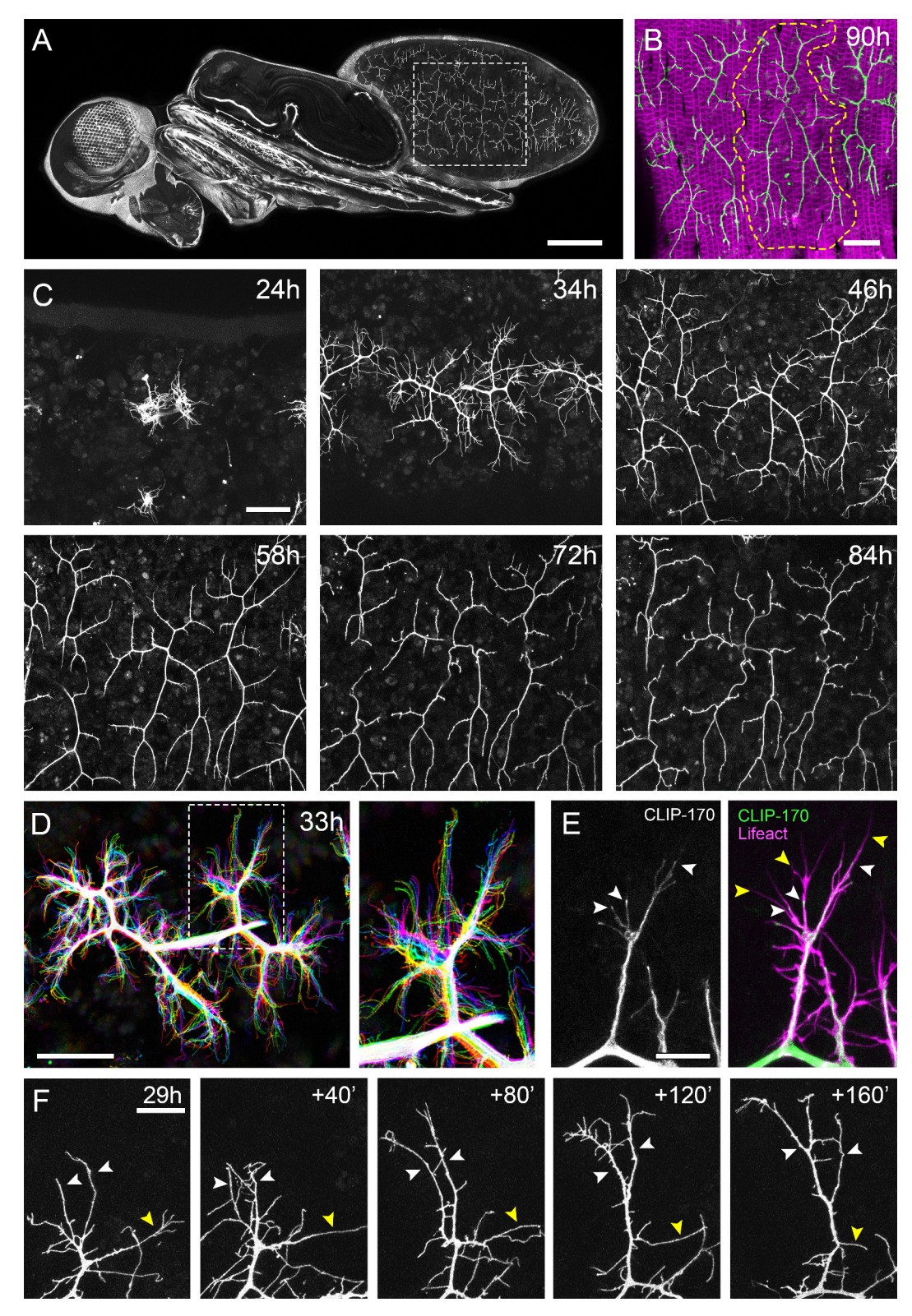

**Figure 1.** The pleural motoneurons show dynamic growth during development. (**A**) A dorso-lateral view of a pupa at 90 hr APF expressing *UAS-myr::GFP* in the pattern of the glutamatergic driver *OK371-GAL4*. Dashed lines demarcate arborisations innervating segments A3 and 4. (**B**) Innervation of the pleural muscles in segments A3 and 4 at 90 hr APF. Muscles labelled with *mCD8::ChRFP* (*Mef2-GAL4*; magenta; **Ranganayakulu et al., 1996**) and motoneurons labelled with *myr::GFP* (*VGlut-LexA*; **Diao et al., 2015**; green). Dashed lines demarcate the anterior arborisation in 4A (A4A). (**C**) A time

*Figure 1 continued on next page*

*Figure 1 continued*

series follows the growth of arborisations in segment A3 from 24 hr to 84 hr APF in 10–14 hr intervals (*VGlut-LexA > myr::GFP*). (D) A 10 min time-lapse (temporally colour coded every 2 min) reveals the dynamic growth of a pair of arborisations in segment A3 at 32 hr APF. (E) Subcellular localisations of cytoskeletal components. Microtubules, revealed by CLIP-170::GFP, are concentrated in branches and dynamically invade nascent branchlets (white arrowheads). Filopodia (yellow arrowheads), are rich in actin, revealed by Lifeact::Ruby, but largely devoid of microtubules. (F) A time series with 40 min intervals shows the growth of a branch at 29 hr APF. White arrowheads indicate filopodia that become stabilised and mature into branches. The yellow arrowhead indicates a branch that retracts into a filopodium. Scale bars: 250 µm (A), 50 µm (B,C,D), 10 µm (E), 20 µm (F).

DOI: https://doi.org/10.7554/eLife.31659.002

filopodia. The two PM-Mns within each hemisegment segregate into anterior and posterior domains and increase in size and complexity until ~72 hr APF. Following this, there is a period of maturation during which varicosities form along all but the most proximal branches. By 84 hr APF the morphology of the arborisations is indistinguishable from that observed at eclosion.

Live imaging at 2 min intervals revealed that branch growth is highly dynamic, involving large numbers of filopodia which extend and retract, continually exploring their local environment (*Figure 1D*). This exploration takes place almost exclusively within the plane of the developing muscles. Throughout this article, we define and refer to filopodia and branches as follows. According to anatomical and molecular criteria, filopodia are considered to be thin (<0.3 µm), motile protrusions with cytoskeletons comprised of parallel F-actin filaments (*Mattila and Lappalainen, 2008*). In contrast, branches are of higher calibre and less dynamic, with cores containing parallel microtubules and noticeably less F-actin. Using Lifeact, a fluorescently tagged F-actin cytoskeletal probe (*Riedl et al., 2008*), we see enrichment in thin 'filopodia-like' structures (*Figure 1E*). In contrast, the plus-end microtubule binding protein CLIP-170 (*Stramer et al., 2010*), that reveals microtubules, is found in what we refer to as branches but is mostly excluded from filopodia. Growing neurites are dynamic and constantly changing, but every effort has been made to be consistent with our classification.

Longer imaging intervals captured the more substantial changes to arbor structure during early growth (*Figure 1F*). This revealed that that the majority of filopodia are transient since large numbers are generated and lost between frames. Despite this, a small number of filopodia persist, pioneering individual branches (white arrowheads). In addition, branches also collapse back into single filopodia (yellow arrowhead), highlighting the instability and continuous remodelling of the arborisations at this early stage (see *Video 1*).

## A close relationship between branching and the distribution of presynaptic components

To explore the growth of PM-Mn axonal arborisations and their muscle targets we simultaneously imaged neurons and muscle progenitors (myoblasts) in the early phases of growth. At 35 hr APF neurons and myoblasts are directly apposed to one another (*Figure 2A* and *Video 2*). Growing axonal branches are enveloped by clusters of myoblasts, while more distal growth cones and filopodia extend over sheets of immature myotubes.

In light of this very close association between the synaptic partners, we asked whether presynaptic components are present in the distal branches at early stages, since previous work in vertebrates has forwarded that nascent synapses play a key role in branch stabilisation. Using an RFP tagged version of Bruchpilot (BRP), a homolog of ELKS/CAST, expressed with *OK371-GAL4* (*Mahr and Aberle, 2006*), we see punctate accumulations at branch nodes and at filopodial bases (*Figure 2B*). An analysis of BRP::RFP puncta in axon terminals of 5 animals staged from 30 hr to 35 hr APF revealed that $85.1 \pm 6.4\%$ (SD) of total BRP puncta were localised at branch points and the bases of filopodia, and $81.6 \pm 4.6\%$ of branch points/bases had a punctum (*Figure 2C*).

To explore the dynamics of BRP::RFP and branching we imaged PM-Mn arborisations at 2 min intervals. This footage showed that BRP::RFP puncta are rapidly recruited to the tips of growing branches where they mark the sites of new filopodia growth (*Figure 2D*). Successive rounds of filopodia extension and stabilisation in this manner produce branches studded with BRP::RFP puncta. Filopodia that extend from these puncta can give rise to new branches (*Figure 2E*).

The dynamics of BRP::RFP puncta speaks to a close relationship between branch/filopodia growth and the distribution of presynaptic proteins (*Figure 2F*). To quantify this we analysed 83 filopodia

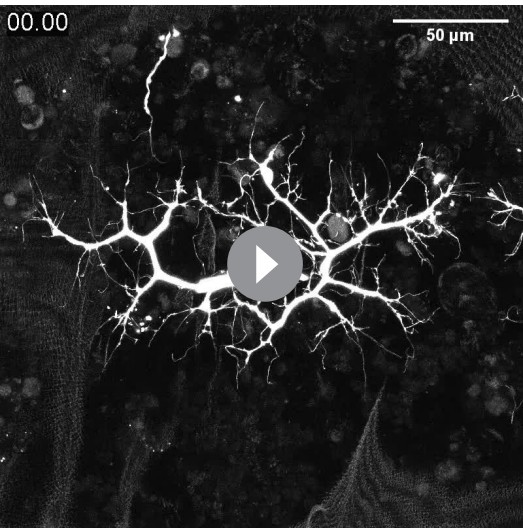

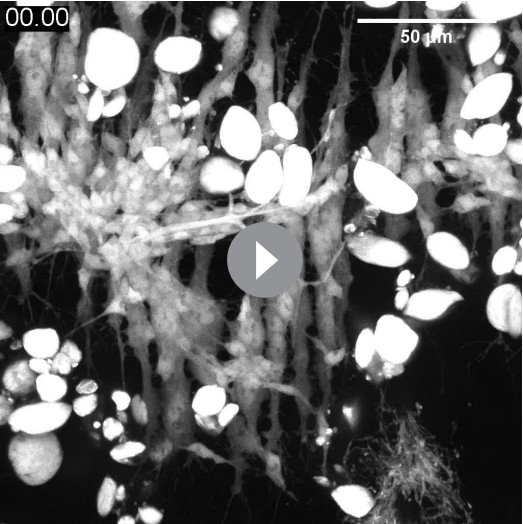

**Video 1.** PM-Mn axon arbor growth is exploratory and highly dynamic. A pair of Pm-Mn axon arborisations growing over 15 hr from 35 hr APF (*OK371-GAL4 > cytoplasmicGFP*). Arbor growth is an exploratory 'trial-and-error' process involving high filopodia turnover and the extension and collapse of larger branch segments. Frames taken at 10 min intervals. Time format: hh:mm.
DOI: https://doi.org/10.7554/eLife.31659.003

**Video 2.** PM-Mn axon arborisations and their synaptic targets grow concurrently and in close association. PM-Mn axon terminals and their muscle partners interact dynamically during early growth (31 hr APF; *OK371-GAL4 + Mef2-GAL4 > cytoplasmicGFP*). A branch projecting into the upper right extends across multiple myotubes via rounds of filopodia extension, stabilisation and maturation. Frames taken at 2 min intervals. Time format: hh:mm.
DOI: https://doi.org/10.7554/eLife.31659.005

generation events from three time-lapse movies of arborisations staged between 30 hr and 33 hr APF. Our analysis found that 36.0 ± 12.6% of filopodia emerged from existing BRP puncta, 48.1 ± 10.7% recruited a punctum within 20 min of their formation and 16.0 ± 11.9% failed to recruit an obvious punctum at all (*Figure 2G*). The frequency of filopodia emerging from or recruiting a BRP:: RFP punctum 71.4 ± 11.0% was significantly greater than a conservative expected frequency of 50% ($X^2$ = 36.4, p<0.0001). In addition, we discovered that 100% of filopodia which failed to recruit puncta to their bases were eliminated within 20 min of emergence (n = 24) (*Figure 2H*). In contrast, although 54.2% of filopodia which originated from, or quickly recruited BRP::RFP puncta were also lost within 20 min, 33.3% survived between 20 and 60 min and a further 12.5% persisted for over an hour (n = 24). Thus, the lifetimes of filopodia associated with puncta at their bases were significantly longer than those without (Mann- two-tailed Whitney U = 73, p<0.0001) (See *Video 3*).

To verify the accuracy of GAL4 driven *BRP::RFP*, we imaged endogenous GFP labelled BRP, since immunohistochemistry is particularly difficult on fragile early pupal body walls (*Figure 2I*). The BRP:: GFP protein trap revealed a punctate distribution at branch points and bases of filopodia, as we had seen with the exogenous reporter. Other GFP tagged protein trap lines that label synaptic vesicle (SV) associated proteins Syt1 and VGlut exhibited less punctate localisations, yet were still clearly concentrated in axon branches and at the bases of filopodia (*Figure 2I*). Interestingly, we never saw these proteins within filopodia.

We wondered whether this node localisation of presynaptic machinery in growing branches is specific to PM-Mns or is more widespread within the fly nervous system. To test this we imaged the output terminals of *Eve* (*Even-skipped*) positive interneurons in the thoracic neuromeres during metamorphosis (24 hr APF) (*Roy et al., 2007*). Here we found punctate distributions of BRP::RFP very similar to those in the PM-Mns, with 79.5 ± 3.7% of puncta located at branch points or bases of filopodia (n = 8 output fields from four individuals; *Figure 2J and K*).

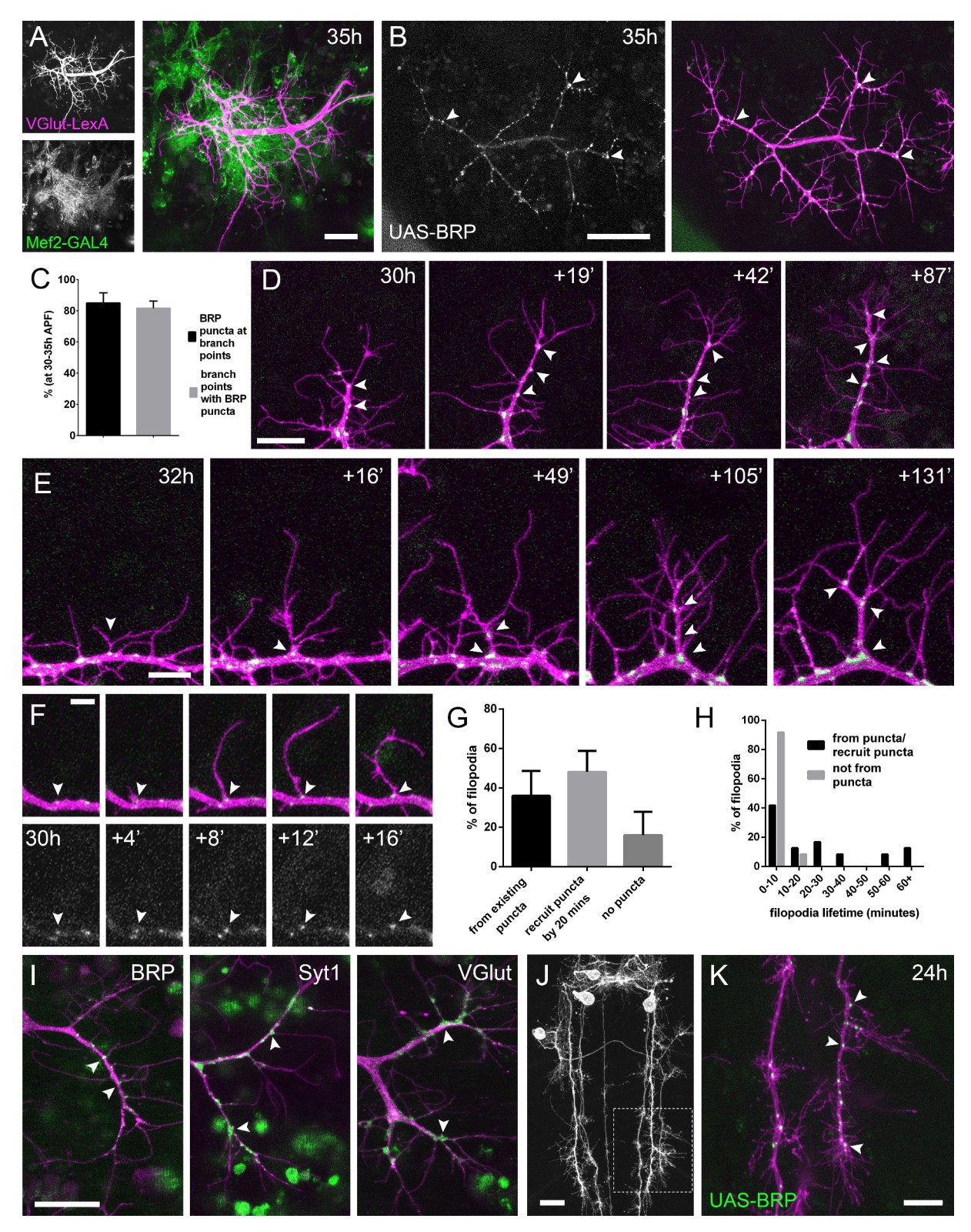

**Figure 2.** The arrival and localisation of presynaptic components is tightly linked with arbor growth. (**A**) Muscles and motoneurons development in close proximity at 35 hr APF. Muscles labelled with *myr::tdTomato* (*Mef2-GAL4*) *and* motoneurons labelled with *myr::GFP* (*VGlut-LexA*). (**B**) A pair of motoneuron axon terminals expressing *BRP::RFP* (green) and *myr::GFP* (magenta) under the control of *OK371-GAL4* at 35 hr APF. BRP::RFP is particularly clear at branch points (arrowheads). (**C**) At stages 30–35 hr APF, 85.05 ± 6.42% of BRP::RFP puncta are localised at branch points or the
*Figure 2 continued on next page*

*Figure 2 continued*

bases of filopodia and 81.62 ± 4.64% of branch points/bases of filopodia host a punctum (n = 5). (D) BRP::RFP puncta are establish following the growth of axon branches. The time series reveals the rapid arrival of BRP::RFP puncta (green) in the growth cone of an extending branch (arrowheads), producing a branch segment studded with puncta. Most BRP::RFP puncta mark sites of filopodia growth. (E) A time series revealing the maturation of a filopodium that originates from a BRP::RFP punctum (arrowhead in first frame) into a branch. Additional BRP::RFP puncta (arrowheads) are rapidly recruited to new branch nodes. (F) Time series shows the emergence of a filopodium from a site marked by a BRP::RFP punctum (arrowhead). (G) The proportions of filopodia which (i) emerge from existing puncta, (ii) recruit a punctum to their base within 20 min and (iii) do not recruit puncta to their bases (83 filopodia from three time-lapse movies at stages 30–33 hr APF). (H) Lifetimes of filopodia which originate from/recruit puncta within 20 min of emergence (21.67 ± 20.30 mins, n = 24) or do not recruit puncta (3.83 ± 2.76 mins, n = 24). Filopodia that originate from puncta/recruit puncta are significantly longer lived than those that do not recruit puncta (Mann-Whitney U = 73, p<0.0001, two-tailed). (I) Protein trap lines reveal the localisation of endogenous BRP, Syt1 and VGlut (MiMIC collection; *Venken et al., 2011*; arrowheads). Arborisations staged between 30 hr and 33 hr APF. (J and K) *Eve + ve* interneuron in T3, at 24 hr APF, (from *RN2-FLP, Tub-FRT-CD2-FRT-GAL4*) with punctate *BRP::RFP* (arrowheads) concentrated at branch points and bases of filopodia in growing output arborisations. Area with dashed lines in (J) is expanded in (K). Bars represent SDs. Scale bars: 20 μm (A,D,I,J), 50 μm (B), 10 μm (E,K), 5 μm (F).

DOI: https://doi.org/10.7554/eLife.31659.004

## PM-Mns elaborate their axonal arborisations without synaptic activity

The localisation and dynamics of presynaptic machineries described above closely mirrors the observations made in *Xenopus* and zebrafish tecta, and suggested a role for synapses and synaptic activity in branch growth during PM-Mn elaboration (*Alsina et al., 2001*; *Niell et al., 2004*; *Meyer and Smith, 2006*; *Ruthazer et al., 2006*).

To explore whether activity plays a role in arborisation growth, we first mapped the development of activity in the PM-Mns using the genetically encoded calcium indicator *GCaMP6m* as a proxy (*Videos 4–6*). Calcium dynamics, recorded as changes in fluorescence, are displayed as heat-registered kymographs at different stages of development (*Figure 3A*). The absence of any overt changes in fluorescence at 32 hr APF indicates that at this stage the motoneurons are electrically inactive. Not until 42 hr APF did we see the first calcium events indicative of membrane depolarisations. These events were isolated and infrequent. The frequency of calcium events increased sharply and by 48 hr APF activity was characterised by episodes of transients in quick succession lasting ~8 min, interspersed with longer periods of quiescence. Approaching the final stages of arbor growth, at 74 hr APF we observed similar patterns of activity, however now bouts contained far greater numbers of events.

To address directly whether arbor growth can take place independently of activity-evoked neurotransmission we silenced neurons by the intravital injection of the sodium channel blocker tetrodotoxin (0.5 mM TTX). Of the eight animals injected between 24 hr and 32 hr APF, six showed no detectable calcium activity at 78 hr APF (*Figure 3B*) and two showed only weak transients restricted to small branch segments. In contrast, buffer injected controls gave expected levels of presynaptic activity (n = 5). At 79 hr APF we found that the arborisations of animals injected with TTX were morphologically indistinguishable from those injected with buffer (*Figure 3C and D*).

In addition to evoked neurotransmission, spontaneous neurotransmitter release, responsible for miniature postsynaptic potentials (mEPSPs), has been shown to play a key role in

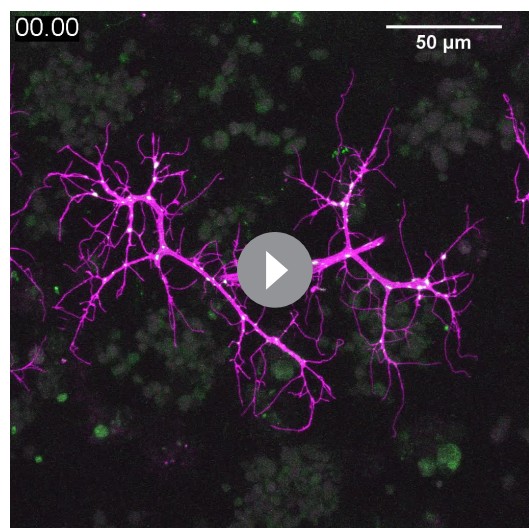

**Video 3.** Arbor growth is associated with distinct localisations of presynaptic proteins. PM-Mn axon arborisations expressing *myr::GFP* and *BRP::RFP* at 32 hr APF (*OK371-GAL4*). BRP::RFP forms puncta which are strongly localised to branch nodes and filopodial bases. Puncta at filopodial bases are correlated with filopodia lifetimes. Frames at 2 min intervals. Time format hh:mm.

DOI: https://doi.org/10.7554/eLife.31659.006

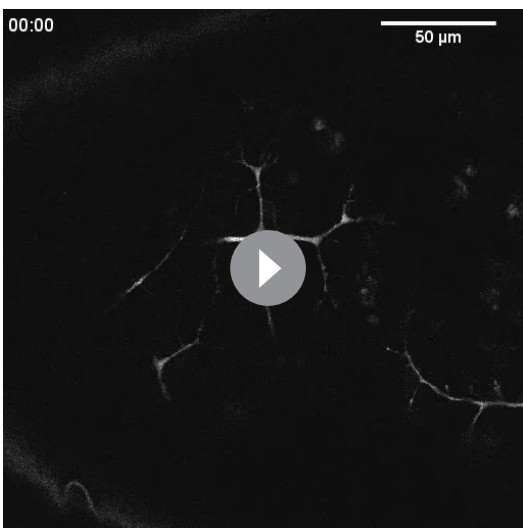

**Video 4.** The development of PM-Mn activity; 32h APF. *GCaMP6m* expressed with *OK371-GAL4* reveals the electrical development of the Pm-MNs. At 32 hr APF the lack of large changes in fluorescence indicates an absence of action potentials. Frames taken at 1 s intervals. Time format mm:ss.

DOI: https://doi.org/10.7554/eLife.31659.008

the growth and development of neurons (*Choi et al., 2014*; *Andreae and Burrone, 2015*). To determine if spontaneous neurotransmission is important for the growth of PM-Mn axonal arborisations we generated PM-Mn clones homozygous mutant for the vesicular glutamate transporter (*VGlut*) using the null allele *Df(2L)VGlut²*. This allele has previously been shown to completely block both evoked and spontaneous glutamatergic neurotransmission (*Daniels et al., 2006*). Antibody staining confirmed a complete loss of VGlut protein in *Df(2L)VGlut²* motoneuron MARCM clones (*Figure 3E*). To assess the effect of removing *VGlut* on arbor growth we performed morphometric analysis on clones of the anterior-most motoneurons of segment A3 (A3-A) in females staged between 80 hr and 90 hr APF (*Figure 3F and G*). No significant difference was found between the area of coverage of *VGlut* null A3-A arborisations and controls (*Figure 3H*), although *VGlut* nulls were marginally greater in total arbor length (*Figure 3I*).

The timeline of neural activity and the *VGlut* null data strongly suggest that synaptic transmission does not play a role during the 'elaborative phase' of growth of the pleural neuromuscular system. To determine when synaptic transmission becomes 'physiologically' possible within this system, we artificially stimulated neurons using the warmth-gated ion channel TRPA1 (*Hamada et al., 2008*), whilst measuring postsynaptic calcium responses in the pleural muscles using GCaMP6m. Before 49 hr APF no calcium events were recorded in the muscles (n = 2; *Figure 3J*). Between 50 hr and 59 hr APF a few large calcium events were recorded in the muscles but in each case these occurred either during the final seconds of stimulation or just after ramping down the temperature (n = 2). In contrast, when stimulating at stages between 68 hr and 72 hr APF we observed robust, rapid and sustained postsynaptic calcium activity (n = 7; *Figure 3J and K*). Between 68–72 hr APF calcium events occurred at significantly greater rates at the permissive temperature (30°C) than at the restrictive temperature (22°C) (6.15 ± 5.60 minute$^{-1}$ vs 0.93 ± 0.81 minute$^{-1}$, Mann-Whitney U = 6.0, n = 7, p=0.02, two-tailed). These data indicate that evoked synaptic transmission does not take place within this system before 60 hr APF.

The lack of impact on growth from removing synaptic transmission caused us to look more closely at the organisation of presynaptic machineries at different times during development (*Figure 3L*). As described previously, at 34 hr APF, BRP::RFP puncta were found almost exclusively at branch points and at the bases of filopodia (white arrowheads). By 48 hr APF, in addition to puncta at branch points, many puncta could also be found along the lengths of branches, between branch nodes (yellow arrowheads). By 58 hr APF all but the most proximal branches were lined with large numbers of BRP::RFP puncta. Finally, at stages 72–77 hr APF, an analysis of the distribution of puncta revealed that just 12.6 ± 5.8%, n = 5 arborisations from five individuals) of puncta were at branch points, the rest being distributed along branch lengths (*Figure 3M*). The majority (89.4 ± 6.7%) of branch points however were still found to have puncta.

In addition to distribution, we used the BRP::GFP protein trap line to look at changes to BRP puncta size (*Figure 3N*). At early stages, puncta were far more heterogenous in diameter than at later stages, measured by a comparison of standard deviations, found to be significant by an F-test of equality of variances ($F_{143,176}$ = 4.092, p<0.0001). We also found that BRP::GFP puncta became significantly smaller over the course of development (*Figure 3O*).

The trend towards smaller, more homogenous puncta later in development highlighted that the puncta we see in early PM-Mn arborisations may not actually represent synapses. To explore this

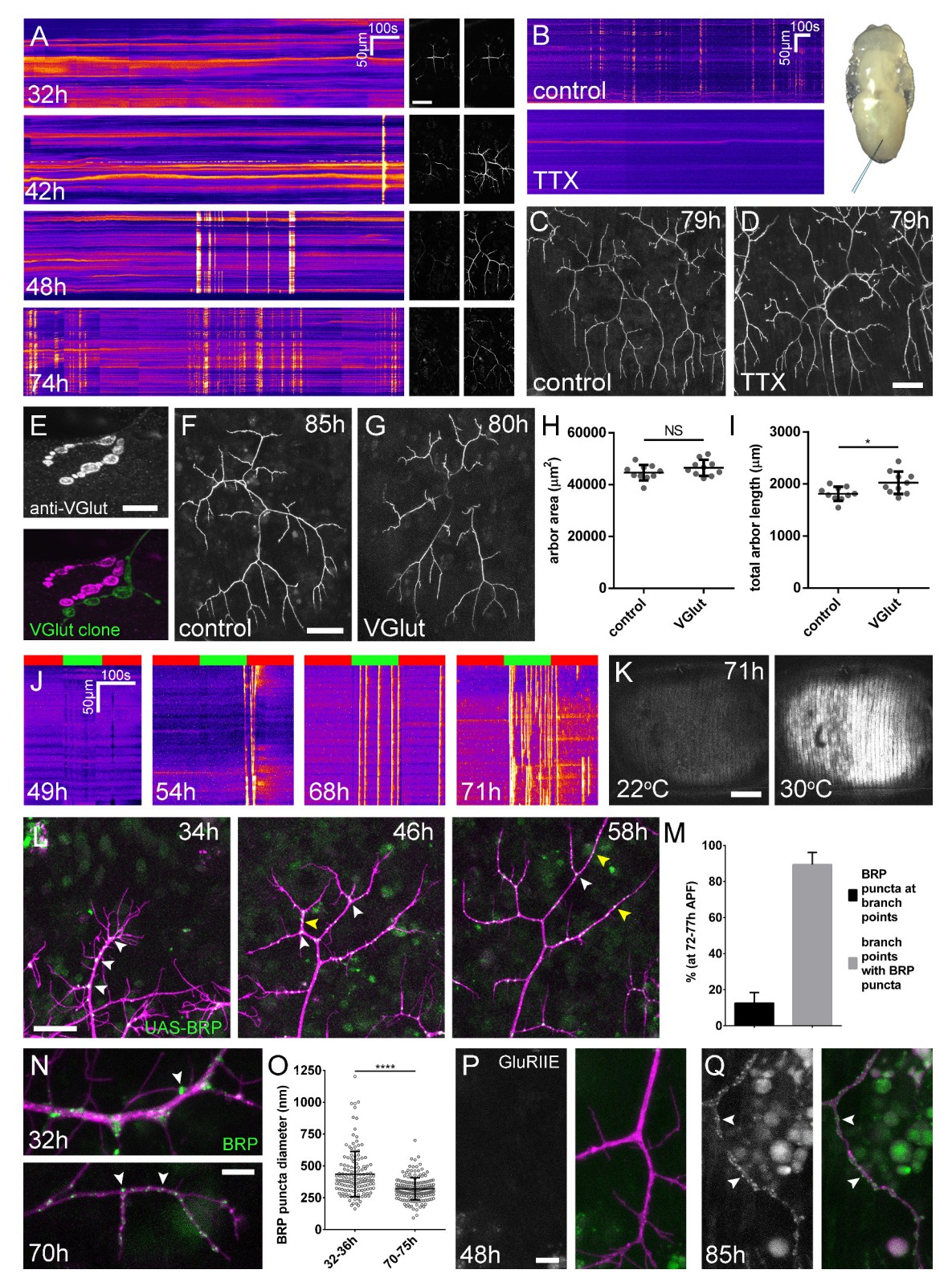

**Figure 3.** Arbor growth takes place without the formation of functional synapses. (**A**) Calcium activity in PM-Mn axon terminals at 32, 42, 48 and 74 hr APF, measured by changes in GCaMP6m fluorescence (Δf) (driven by *OK371-GAL4*). Sequential images show examples of Δfs displayed as heat registered kymographs (**B**) GCaMP6m Δfs in PM-Mn axon terminals at 78 hr APF in a pupa injected with a control solution of PBS (top) and a solution of 0.5 mM TTX in PBS (bottom) into the abdomen of a pupa (inset). (**C**) Arborisations in segment A3 (*OK371-GAL4 > myr::GFP*) at 79 hr APF following

*Figure 3 continued on next page*

*Figure 3 continued*

injection at 32 hr APF with a PBS solution and (**D**) solution containing TTX. (**E**) Motoneuron axon terminal of a *VGlut* null MARCM clone (green) alongside a *VGlut* heterozygous terminal in L3 larva stained with an antibody against VGlut (magenta). (**F**) The anterior-most arborisation in segment A3 (A3–A) of a heterozygous *VGlut* control expressing *myr::GFP* and *mCD8::GFP* (*VGlut*[NMJX]-*GAL4*) at 85 hr APF (**G**) an equivalent arborisation in *VGlut* null MARCM clone at 80 hr APF. (**H and I**) Arborisations of *VGlut* null A3-A MARCM clones do not differ significantly from controls in arbor area (controls: $44643 \pm 2982 \ \mu m^2$, *VGlut*: $46518 \pm 3083 \ \mu m^2$, $n_{1,2} = 10$, $t(18) = 1.38$, p=0.18, t-test, two-tailed). Marginally greater total arbor lengths (controls: $1812 \pm 134 \ \mu m$, *VGlut*: $2024 \pm 216 \ \mu m$, $n_{1,2} = 10$, $t(18) = 2.64$, p=0.02, t-test, two-tailed). (**J**) Muscle GCaMP6m (*Mef2-GAL4*) $\Delta f$ in response to motoneuron activation at 49 hr, 54 hr, 68 hr and 71 hr APF, using the warmth-gated ion channel TRPA1 (*VGlut-LexA > dTRPA1*). Red bars indicate time at the restrictive temperature (22°C), green bars indicate time at the permissive temperature (30°C). (**K**) Images show muscle GCaMP6m $\Delta fs$ before and after activation of motoneurons with dTRPA1 at 71 hr APF. (**L**) Organisation of BRP::RFP puncta through development. Time series of the same arbor segment at 34 hr, 48 hr and 56 hr APF shows a shift from puncta at branch points (white arrowheads) to a distribution along branch lengths (yellow arrowheads). (**M**) Between 72 hr and 77 hr APF, $12.55 \pm 5.79\%$ of total BRP::RFP puncta are found at branch points/bases of filopodia, yet the majority ($89.38 \pm 6.73\%$) of branch points/bases of filopodia host puncta (n = 5). (**N**) Size and distribution of endogenous, GFP labelled BRP. The puncta of BRP (indicated by arrowheads) are larger and more heterogenous in diameter during early arbor growth (32 hr APF) than at later stages (70hAPF), when the major phase of outgrowth has ceased (*OK371-GAL4 > mCD8::ChRFP*). (**O**) Diameters of endogenous BRP::GFP puncta measured as the full width at half maximum of peaks in fluorescence are significantly greater at 32 hr APF ($435.8 \pm 177.8$ nm, n = 144) than at 72 hr APF ($319.6 \pm 87.9$ nm, n = 177) (Mann-Whitney U = 6935, p<0.0001, two-tailed). (**P**) Localisation of GluRIIE at 48 hr APF. This GFP tagged version of *GluRIIE* (FlyFos; *Sarov et al., 2016*) driven under the control of the native transcriptional unit cannot be seen in the postsynaptic membrane before or at 48 hr APF. (*OK371-GAL4 > mCD8::ChRFP*). (**Q**) At 85 hr APF conspicuous GluRIIE::GFP clusters (arrowheads) are found along the axon terminals. Bars represent SDs. Scale bars: 50 μm (**A, C,D,F,G**), 10 μm (**E**), 100 μm (**K**), 25 μm (**L**), 5 μm (**N,P,Q**).

DOI: https://doi.org/10.7554/eLife.31659.007

possibility we looked to see when glutamate receptors first appear in the muscles. No detectable GluRIIE was evident at 48 hr APF (*Figure 3P*) yet by 85 hr APF clear clusters of GluRIIE were found apposed to the branches (*Figure 3Q*). Thus, the changes in the organisation of synaptic machineries over the course of development suggest that the early accumulations of presynaptic components do not represent differentiated synapses.

## A role for Neurexin-Neuroligin 1 signalling in PM-Mn arborisation growth

Neurexin-Neuroligin signalling is capable of driving synapse formation in many contexts, and in the *Xenopus* tectum regulates arbor growth in an activity dependent manner (*Chen et al., 2010*). Though our data rules out a key role for activity in PM-Mn axonal arbor growth, we next sought to test whether these synaptic adhesion proteins may be important.

To generate homozygous mutants, we used the null alleles *Nlg1*[ex2.3], *Nlg1*[1960] (*Banovic et al., 2010*) and *Nrx*[241] (*Li et al., 2007*) in combination with chromosomal deficiencies. These combinations are viable into adulthood. At late pupal stages we found that both *Nlg1* and *Nrx* nulls show comparable defects in arbor morphology, with large reductions in coverage relative to wild-type controls (*Figure 4A and B*).

To quantify these effects, we used morphometric methods of analysis. *Nlg1* and *Nrx* null arborisations showed significant reductions in area of coverage and total arbor length compared to wild-type controls (*Figure 4C and D*). We found no differences between total branch numbers (*Figure 4E*). *Nrx* and *Nlg1* null arborisations did not differ significantly from each other in any of these measurements. To determine if arbor complexity was different in nulls we used the Strahler method of branch ordering (*Figure 4F*). With this we found no differences in the maximum number of branch orders. Furthermore, the total number of lowest order (terminal) branches did not differ significantly between the groups; a trend which continued for subsequent orders, with all conditions having remarkably similar numbers of branches at every level.

The morphology of late stage *Nlg1* and *Nrx* nulls points toward a requirement for these proteins for growth, but does not tell us *when* they are required. To address the timing of requirement, we compared *Nlg1* null arborisations with those of wild-type controls at 30–36 hr APF. PM-Mns in *Nlg1* nulls at this stage generate similar numbers of very dynamic filopodia compared to controls. A clear difference however was the far greater 'bendiness' – or tortuosity – of *Nlg1* null branches (*Figure 4G*). To evaluate this, we scored primary and secondary branches of *Nlg1* null and control arborisations staged between 30 hr and 36 hr APF using an index of bendiness (*Figure 4H*). This was calculated from the percentage difference between the actual length of branches and the

straight-line distance between their nodes. *Nlg1* null branches were found to be significantly less straight than those of the controls (Mann-Whitney U = 441, p<0.0001, two-tailed). This early phenotype demonstrates a requirement for Nlg1 signalling during the very initial phases of pleural neuromuscular development.

Although the full mutants showed robust and consistent phenotypes, there is limitation on what these can tell us. They cannot, for example, reveal whether the phenotype is due to a failure of 'local interactions' between the developing synaptic partners. To address this we generated FLEX-based, FlpStop tools (*Fisher et al., 2017*) which allow conditional disruption of endogenous Nlg1 expression in a clonal fashion (*Figure 5A and B*).

Using a MiMIC insertion within the third coding intron of *Nlg1* we generated FlpStop lines capable either of mosaically rescuing *Nlg1* in a mutant background, or mosaically disrupting *Nlg1* in a wild-type background. For complete gene disruption, FlpStop lines were used in heterozygous condition with a deficiency covering *Nlg1*. To test the ability of the non-disrupting (ND) orientation to disrupt Nlg1 expression upon cassette inversion, we used *hsFlp* to induce large numbers of Disrupting (D-lock) clones. These dis-

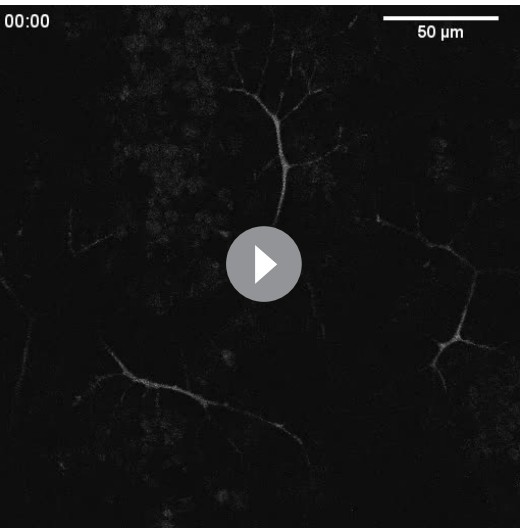

**Video 5.** The development of PM-Mn activity; 43h APF. *GCaMP6m* expressed with *OK371-GAL4* reveals the electrical development of the Pm-MNs. By 43 hr APF calcium activity is characterised by large changes in fluorescence indicative of membrane depolarisations. Frames taken at 1 s intervals. Time format mm:ss.
DOI: https://doi.org/10.7554/eLife.31659.009

played a very similar arbor growth phenotype to *Nlg1* nulls (*Figure 5D*). To test the ability of the disrupting orientation (D) to be converted into a non-disrupting allele (ND-lock), a germ-line inverted stock was generated. In this case, arbor growth was rescued to a near wild-type phenotype (*Figure 5E*).

To test the local action of Nlg1, we induced small numbers of FlpStop muscle precursor clones. This generated fibres containing varying numbers of Nlg1 +ve nuclei (*Figure 5F and H*). When viewed at stages between 70 hr and 85 hr APF, the organisation of arborisations, relative to the pattern of clonal muscle fibres, suggested a local role for Nlg1 mediated adhesion in branch growth. Using the allele in the initially non-disrupting orientation, terminals growing on non-clonal/low level clonal muscle appeared phenotypically closer to wild-types, whereas branches of neurons growing on muscle with more D-lock clones displayed phenotypes comparable to complete Nlg1 nulls (see *Figure 4B*; n = 8 muscle mosaics; *Figure 5G*). Using the allele in the initially disrupting orientation the reverse situation was observed; branches growing across ND-lock clonal myotubes, in which Nlg1 expression was rescued, displayed more wild-type-like morphologies than those growing on non-clonal fibres (n = 7 muscle mosaics; *Figure 5I*).

## Dynamic complexes of 'synaptic' adhesion proteins stabilise filopodia and drive branch growth

One prediction from these mutant data is that Nlg1 might 'prepattern' PM neuromuscular junctions, as is seen with acetylcholine receptors in zebrafish somatic muscles (*Panzer et al., 2006*). To explore this idea we looked at the localisation of postsynaptic Nlg1 using GFP labelled Nlg1 (*Banovic et al., 2010*) expressed under the control of *Mef2-GAL4*. At 35 hr APF Nlg1::GFP faintly labels the entire postsynaptic membrane, but forms strong puncta only at sites that directly appose the axon terminals (*Figure 6A*). Higher magnifications showed a concentration of puncta on growing axonal branches, particularly at branch points and at the very tips of exploratory filopodia (*Figure 6B* and *Video 7*).

To ask how Nlg1::GFP is recruited to branches we recorded growth at 5 min intervals at 35 hr APF (*Figure 6C*). In this footage we found that Nlg1::GFP puncta are recruited directly onto

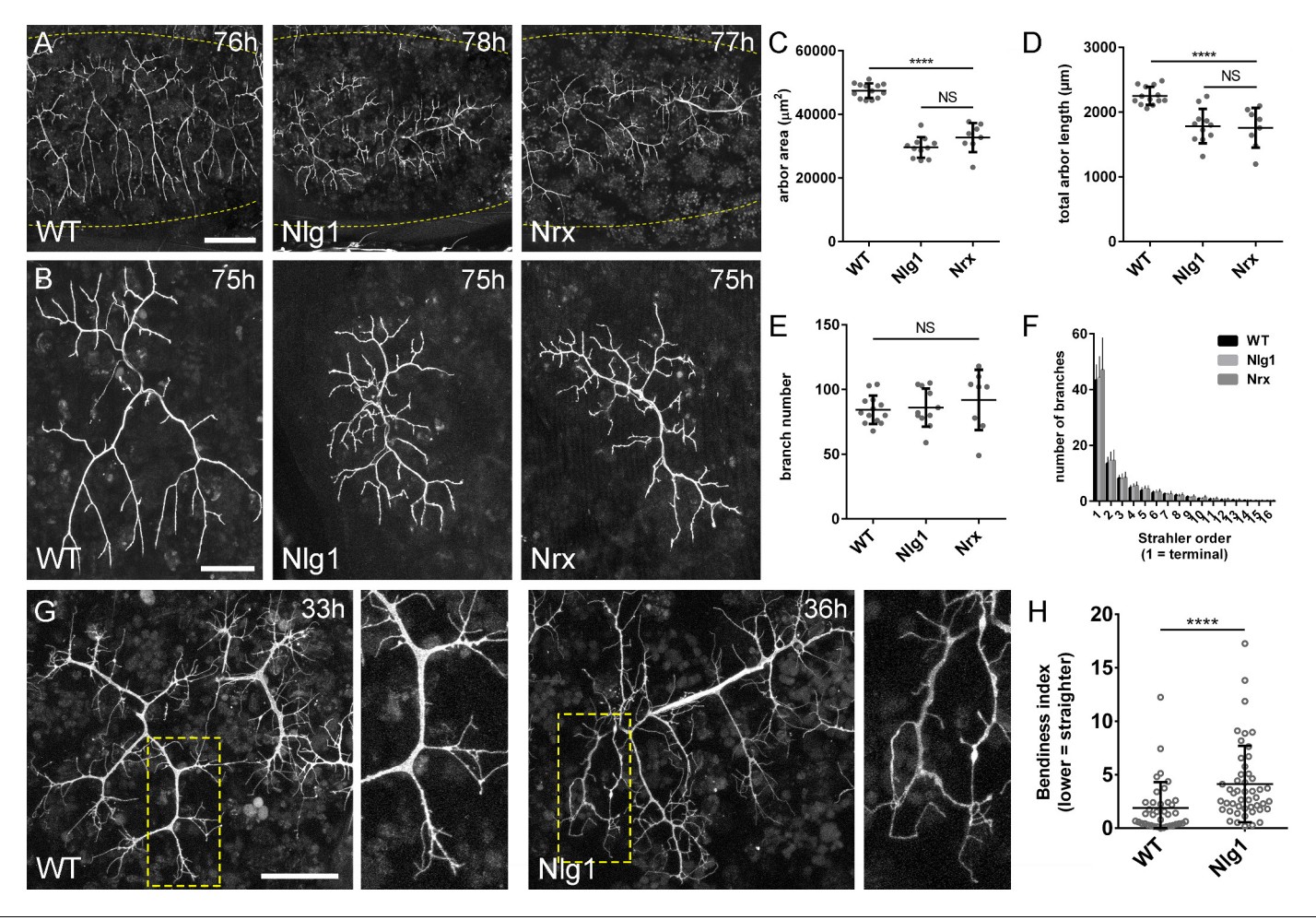

**Figure 4.** *Neurexin* and *Neuroligin 1* are required during arbor growth. (**A**) Arborisations of PM-Mn axons in segments A3-A5 in a wild-type control, an *Nlg1* null and an *Nrx* null staged between 76 hr and 78 hr APF (*VGlut-LexA > myr::GFP*). Yellow dots show the outline of epidermis (**B**) Anterior-most arborisations in segment A3 (A3–A) at stages between 75 hr and 85 hr APF. Arborisations belonging to other motoneurons alongside have been removed using image processing. (**C–F**) Morphometric analysis of A3-A arborisations of *Nlg1* nulls (n = 11), *Nrx* nulls (n = 8) and wild-type controls (n = 13) staged from 75 hr to 85 hr APF. (**C**) *Nlg1* and *Nrx* nulls cover significantly lower territories (controls: 47445 ± 2298 µm$^2$, *Nlg1*: 29616 ± 3259 µm$^2$, *Nrx*: 32696 ± 4565 µm$^2$. *Nlg1* vs controls: t(22) = 15.67, p<0.0001. *Nrx* vs controls: t(19) = 9.89, p<0.0001. t-tests, two-tailed) (**D**) Total PM-Mn arbor lengths controls: 2251 ± 140 µm, *Nlg1*: 1785 ± 265 µm, *Nrx*: 1758 ± 306 µm. *Nlg1* vs controls: t(22) = 5.51, p<0.0001. *Nrx* vs controls: t(19) = 5.07, p<0.0001. t-tests, two-tailed), are significantly different. (**E**) Total number of PM-Mn branch numbers are not significant (controls: 84.38 ± 10.97, *Nlg1*: 86.00 ± 14.74, *Nrx*: 91.88 ± 23.27. *Nlg1* vs controls: t(22) = 0.31, p=0.76. *Nrx* vs controls: t(19) = 1.00, p=0.33. t-tests, two-tailed). (**F**) *Nlg1* nulls, *Nrx* nulls and controls were no different in their topological organisation. Nulls did not differ from controls in their total number of highest order (terminal) branches (controls: 43.31 ± 5.59, *Nlg1*: 44.27 ± 7.56, *Nrx*: 47.00 ± 11.50. *Nlg1* vs controls: t(22) = 0.36, p=0.72. *Nrx* vs controls: t(19) = 0.99, p=0.33. t-tests, two-tailed) or in their total number of orders (controls: 11.62 ± 1.94, *Nlg1*: 11.27 ± 2.15, *Nrx*: 11.88 ± 2.36. *Nlg1* vs controls: t(22) = 0.41, p=0.67. *Nrx* vs controls: t(19) = 0.27, p=0.79. t-tests, two-tailed). (**G**) At early stages of outgrowth *Nlg1* null arborisations possess a number of branches that grown precociously, yet are more tortuous and with fewer side branches than wild-types. (**H**) Branch bendiness of PM–Mns arbors between 32 hr and 36 hr APF. Bendiness was measured as the percentage difference between the actual length and the straight-line length of primary (terminal) and secondary branches. Branches of *Nlg1* nulls are less straight (4.13 ± 3.57%, n = 48) than controls (1.89 ± 2.41%, n = 38) at stages between 32 hr and 36 hr APF (Mann-Whitney U = 441, p<0.0001, two-tailed). Bars represent SDs. Scale bars: 100 µm (**A**), 50 µm (**B,G**).

DOI: https://doi.org/10.7554/eLife.31659.011

filopodia and at the tips of branches. As a result, branch growth occurs as a highly coordinated sequence where the arrival of Nlg1::GFP on exploratory processes precedes their stabilisation and maturation into stable branches in an iterative manner.

The dynamic recruitment of Nlg1::GFP puncta to growing branches and filopodia points to an important structural role for Nlg1 during early arbor growth. To look at the relationship between

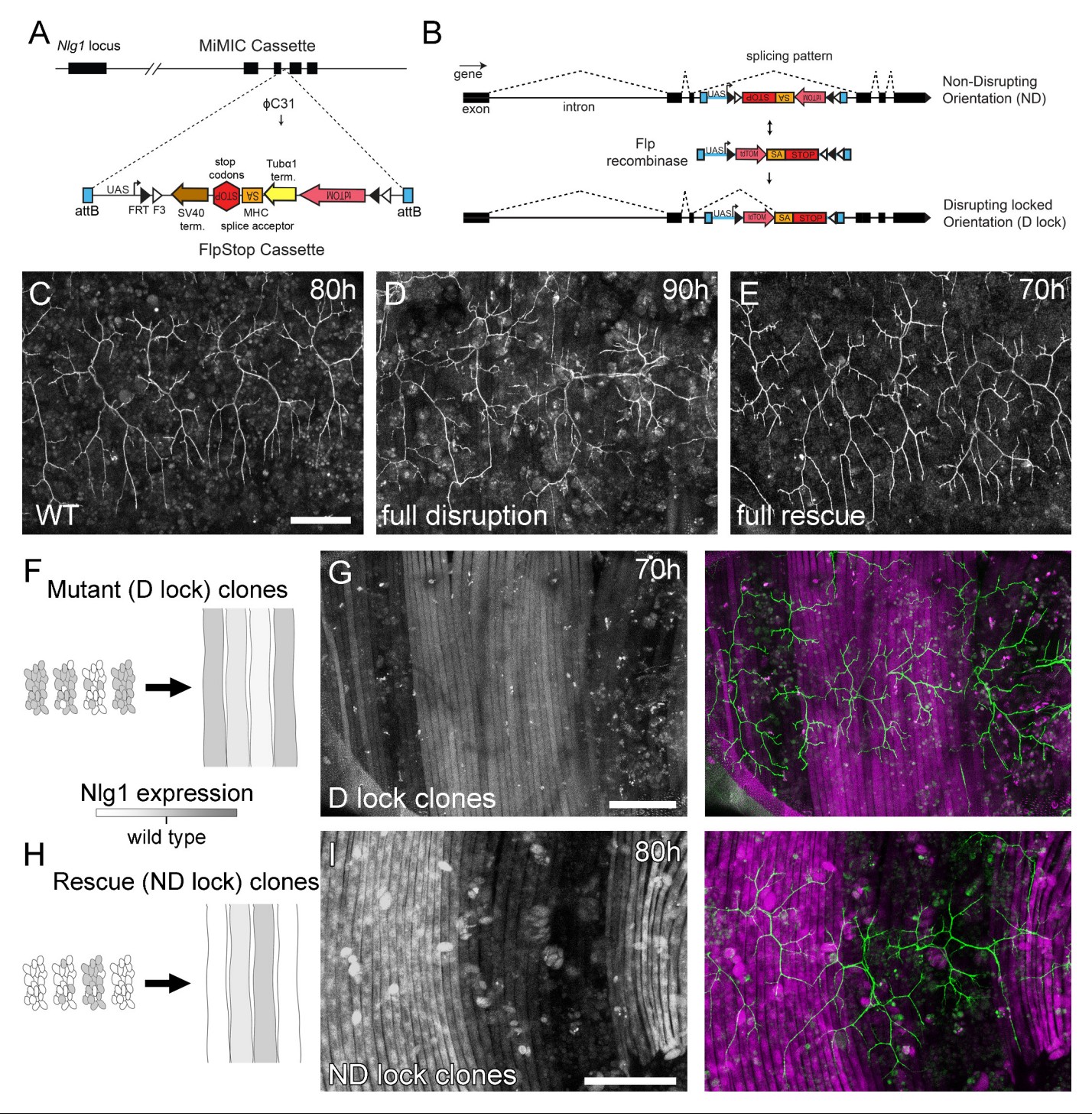

**Figure 5.** Mosaic analysis shows a local role for endogenous *Neuroligin 1* during arbor growth. (**A**) A schematic shows the directed insertion of the FlpStop construct into the third coding intron of the *Nlg1* genomic locus by recombinase mediated cassette exchange (RMCE) with a MiMIC integration site. (**B**) A schematic shows the mechanism of FlpStop action for the conditional disruption of endogenous *Nlg1*. In the initially non-disrupting (ND) orientation the splice acceptor and stop codons are on the non-coding strand and thus do not effect gene function. Upon inversion by FLP recombinase the splice acceptor and stop codons are brought into frame on the coding strand, resulting in disrupted gene expression. The cassette is locked into this disrupting (D-lock) orientation by a FLEx switch (*Schnütgen et al., 2003*). In addition, inversion brings the coding sequence for tdTomato into proximity with a UAS sequence, enabling GAL4 driven expression. FlpStop lines were also generated in the initially disrupting (**D**) orientation, which allows rescue of *Nlg1* (ND-lock clones) and 'turns on' tdTomato (Schematic not shown). (**C**) Arborisations at 80 hr APF in a wild-type control (*VGlut-LexA > myr::GFP*). (**D**) Complete induction of *Nlg1* D-lock (disrupting) clones using *hsFlp* and a long heat-shock protocols produces

*Figure 5 continued on next page*

Figure 5 continued

arbor growth defects comparable to *Nlg1* mutants. (**E**) Germline ND-lock rescues arbor growth to a near wild-type phenotype. (**F**) Schematic shows the formation of *Nlg1* deficient fibres from the fusion of D-lock clonal myoblasts induced using *hsFlp* and a short heat shock at larval L3 stage. (**G**) Arbor growth (visualised with *VGlut-LexA*) onto regions of D-lock muscle clones (*Mef2-GAL4*) is disrupted, whereas arbor growth on non-clonal regions is close to wild-type. (**H**) Model shows the rescue of *Nlg1* expression in clonal fibres formed from the fusion of ND-lock clonal myoblasts induced using *hsFlp* and a short heat shock at larval L3 stage. (**I**) Arbor growth onto ND-lock muscle clones is close to wild-type, whereas growth on non-clonal regions is disrupted. Scale bars: 100 μm (**C,D,E,G,I**).

DOI: https://doi.org/10.7554/eLife.31659.012

Nlg1::GFP puncta and filopodia/branch dynamics, we took higher frequency time-lapses of arborisations between 30 hr and 35 hr APF. Shown in the series in *Figure 6D*, Nlg1::GFP puncta in apposition to growth cones and filopodia were often found to be stable for many minutes. Furthermore, puncta apposed to filopodia appeared to correlate with filopodia longevity (white arrowheads), and regularly marked the limits of filopodial retraction (yellow arrowhead). To assess this relationship, we analysed the lifetimes of filopodia with and without Nlg1::puncta in time-lapses from seven individuals. Shown in the graph in *Figure 6E*, 55.1% of filopodia not apposed to puncta (n = 49 filopodia) were lost within 10 min, 22.5% were lost within 20 min and a further 22.5% lasted longer than the duration of the movies. On the other hand, only 4.4% (2 filopodia) of the population apposed to puncta (n = 45 filopodia) were lost within 10 min, 2.2% (1 filopodium) were lost within 20 min and 93.3% survived for longer than the duration of the recordings. Thus, filopodia tipped by Nlg1::GFP puncta were significantly longer lived than filopodia not bearing puncta (*Figure 6E*).

The major trans-synaptic binding partners of Neuroligins are the Neurexins. A natural prediction from this would be that presynaptic Nrx mirrors the postsynaptic distribution of Nlg1::GFP. Using GFP tagged *Nrx1* (hereon termed *Nrx*) (*Banovic et al., 2010*) expressed in motoneurons we observed a punctate distribution in growing branches, particularly at points of filopodia growth and at filopodial tips, much like Nlg1::GFP (*Figure 6F* and *Video 8*). The low signal and rapid bleaching of Nrx::GFP made it difficult to follow Nrx dynamics in vivo over longer periods.

Alongside Nrx we also looked at two key players in presynaptic development; Syd-1 and Liprin-α. Liprin-α (Lar interacting protein) is a scaffolding protein that is one of the first components recruited to trans-synaptic Nrx-Nlg1 complexes (*Owald et al., 2012*). Liprin-α::GFP (*Fouquet et al., 2009*) was expressed in motoneurons with *OK371-GAL4*. Much like Nrx::GFP, Liprin-α::GFP forms distinct puncta at branch terminals and within filopodia (*Figure 7A*). Similarly to postsynaptic Nlg1::GFP, we found that presynaptic Liprin-α::GFP puncta often mark the limits of filopodial retraction (*Figure 7B*) and appeared to correlate with filopodial stability (see *Video 9*). Indeed, filopodia bearing Liprinα::GFP puncta were significantly longer lived than those not (*Figure 7C*). It appears that Liprin-α::GFP coalesces into puncta directly on filopodia (*Figure 7D*). To ask if Liprin-α::GFP puncta mark adhesion complexes we looked at the localisation of another known interactor, *Syd1*. Syd1:: GFP (*Owald et al., 2010*), like Liprin-α::GFP, forms puncta which localise within growing branch terminals, including to the tips of filopodia (data not shown). To see if these two proteins colocalise to the same sites we expressed *Liprin-α::GFP* together with Strawberry tagged *Syd1* (*Syd1::Straw*) (*Owald et al., 2010*). The large majority of puncta of each protein were coincident, including those at the tips of filopodia (*Figure 7E*). In contrast, when *Liprin-α::GFP* and *BRP::RFP* were expressed together there was very little co-localisation at branch points and never at filopodia tips (*Figure 7F*). Only much later in development do BRP and Liprin-α become arranged like at active zones at the larval NMJ (*Fouquet et al., 2009*), with clusters of Liprin-α abutting the edges of BRP puncta (*Figure 7G*).

To explore if Liprin-α localisation requires Nlg1, we looked at Liprin-α::GFP in *Nlg1* nulls (*Figure 7H*). Without Nlg1 we found that Liprin-α::GFP still localises to growing axon terminals, including to the tips of some filopodia. On closer inspection however, we found that Liprin-α::GFP puncta are largely absent from the unusually long and unbranched terminal branches, typical of growing Nlg1 null arborisations (white arrowheads). When followed, these branches invariably collapse back or fail to grow any further (n = 4 arborisations from four individuals), resulting in the characteristically stunted appearance of *Nlg1* null arborisations.

Finally, to determine the subcellular distribution of Liprin-α::GFP in the central nervous system we looked at *Eve* + ve interneurons. Like in growing axon terminals of PM-Mn neurons, Liprin-α::GFP is

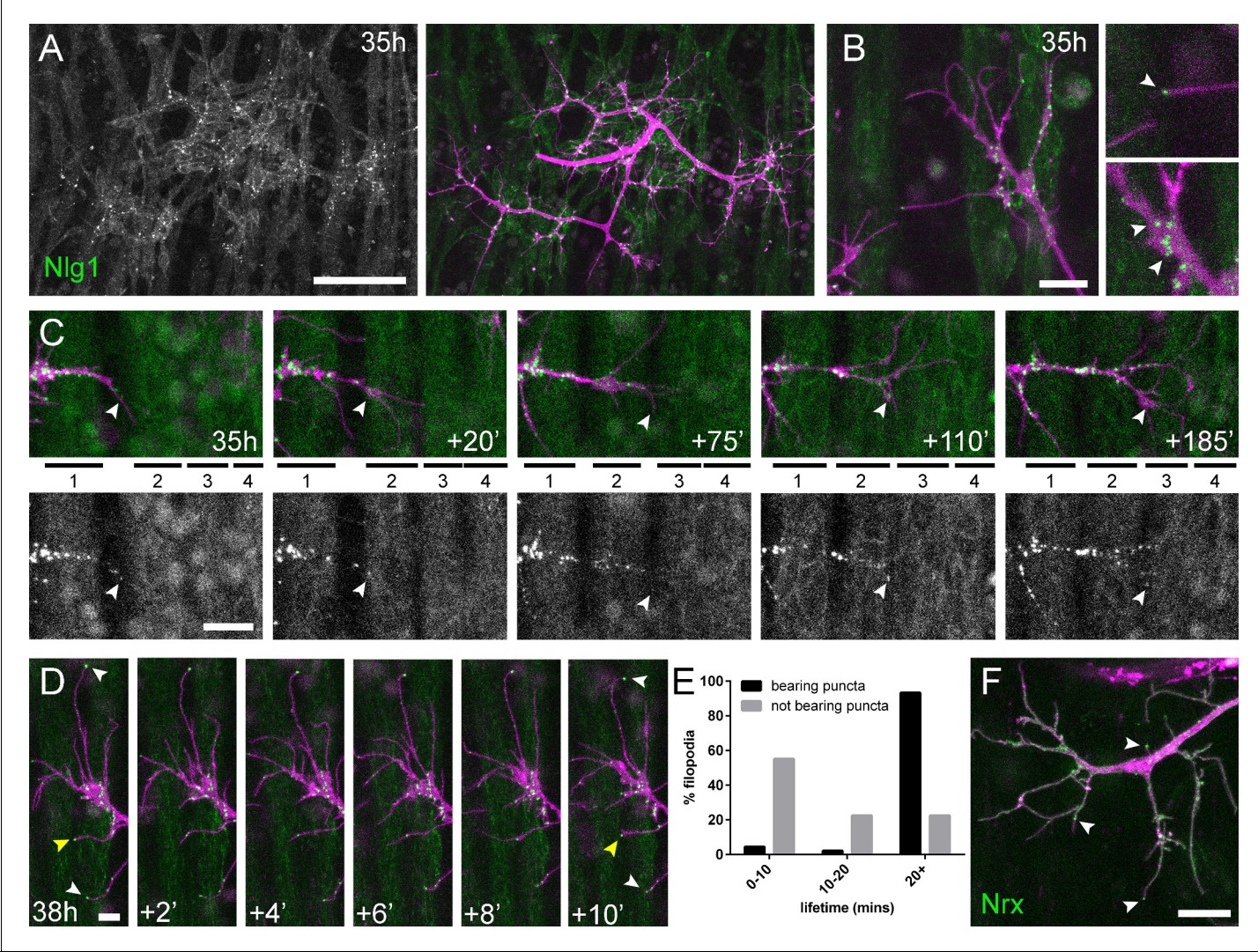

**Figure 6.** Neuritic Adhesion Complexes (NACs) containing *Neurexin* and *Neuroligin 1* stabilise filopodia during arbor growth. (**A**) *Nlg1::GFP* (*Mef2-GAL4*; green) expressed in muscles at 35 hr APF forms puncta on the postsynaptic membrane that are arranged exclusively in apposition with the axon terminals (*VGlut-LexA > myr::tdTomato*; magenta). (**B**) Higher magnifications show the concentration of Nlg1::GFP puncta at sites of branch growth and at filopodia tips. (**C**) Rounds of filopodia extension and stabilisation are concomitant with the recruitment of Nlg1::GFP puncta (white arrowheads) to their tips. Time series shows the extension of a branch across a muscle field at 35 hr APF (myotubes numbered from anterior to posterior). (**D**) Detail of relationship between the dynamics of filopodia and Nlg1::GFP puncta. Puncta (white arrowheads) mark the tips of filopodia that persist. Yellow arrowhead indicates a punctum marking the limit of filopodial retraction. (**E**) Relationship between lifetimes of filopodia and Nlg1::GFP puncta. Histogram showing lifetimes of filopodia with Nlg1::GFP puncta at their tips (n = 45) and without puncta at their tips (n = 49). Filopodia bearing Nlg1::GFP puncta were significantly longer lived than those without puncta (with puncta: 19.36 ± 2.82 mins, without puncta: 10.73 ± 6.88 mins, Mann-Whitney U = 305.5, p<0.0001, two-tailed). (**F**) Localisation of *Nrx::GFP* (*OK371-GAL4*; green) in PM-Mn growing axonal arborisation at 35 hr APF (*mCD8::ChRFP*; magenta). Arrowheads indicate puncta positioned on filopodia. Bars represent SDs. Scale bars: 50 µm (A), 10 µm (B,F), 20 µm (C), 5 µm (D).
DOI: https://doi.org/10.7554/eLife.31659.013

localised to the tips and along the lengths of filopodia on the growing axonal output arborisations (*Figure 7I*).

## Nlg1-based adhesion complexes can direct growth in a tropic manner

The relationship between Nlg1::GFP puncta and filopodial dynamics points to a role for Nlg1 in the growth of axonal arborisations by providing adhesive stability to branches and filopodia. A direct prediction from this would be that if we manipulated postsynaptic levels of Nlg1 at early stages we

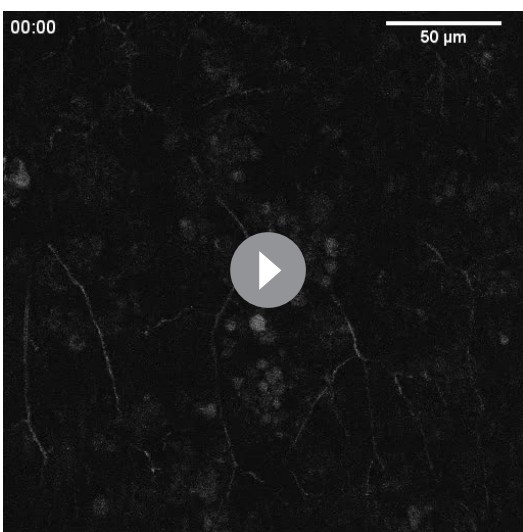

**Video 6.** The development of PM-Mn activity; 78h APF. *GCaMP6m* expressed with *OK371-GAL4* reveals the electrical development of the Pm-MNs. By 78 hr APF activity is defined by bouts of high frequency transients. Frames taken at 1 s intervals. Time format mm:ss.

DOI: https://doi.org/10.7554/eLife.31659.010

would see corresponding changes in PM-Mn growth. To explore this we used *UAS-Nlg^untagged*, which is known to be expressed at high levels and causes a strong larval NMJ phenotype (*Banovic et al., 2010*).

Elevated postsynaptic Nlg1 levels resulted in compacted axon terminals at 33–35 hr APF, with distal branches transformed into flattened growth cones with many filopodia (*Figure 8A and B*). This significant effect on growth led to late stage arborisations with reduced branch length, territory and complexity (*Figure 8C and D*). These branches also maintained greater than usual numbers of filopodia into the later stages of development.

Although full postsynaptic overexpression of *Nlg1^untagged* revealed that axon terminals are sensitive to Nlg1 levels from very early stages, to explore how different levels of Nlg1 signalling impacts branch growth at a local level we developed a clonal technique for generating patterns of expression much like a 'Bonhoeffer stripe' assay (*Walter et al., 1987*). With this 'stripe assay' we present growing motoneurons with myotubes expressing different levels of Nlg1^untagged on which to grow upon (*Figure 8E*). At early stages (35 hr APF), PM-Mn growth appeared to be directed onto myoblast clusters/developing myotubes that strongly expressed the *Nlg1^untagged* (*Figure 8F*). By later stages (70 hr APF), branches in contact with these strongly expressing clonal fibres displayed the same hyper-stabilisation phenotype seen with full muscle expression. In contrast, branches from the same neuron in contact with low to non-expressing fibres grew as normal, demonstrating unequivocally that Nlg1 impacts branch growth via local mechanisms (*Figure 8G*). Although branches contacting highly expressing fibres had reduced growth and branching, they could sometimes be seen to elaborate along a clonal fibre, suggesting that a 'tropic' mode of growth is at play.

To further investigate Nlg1 signalling on branch growth we took advantage of the organisation of the sensory nervous system in the abdominal body wall. As shown in *Figure 8H*, the class IV dendritic arborisation sensory neuron, v'ada, elaborates on the epidermis. The peripheral neurites of v'ada are normally separated from the motor axon branches by the dorsoventral pleural muscles, which in turn are innervated by the PM-Mns on their internal surface. During early pupal development, both motor and sensory arborisations are found in very close proximity. We predicted that if we expressed Nlg1 in these sensory neurons, signalling would direct the growth of the PM-Mn arborisations into a novel territory and encourage them to make connections with these normally 'asynaptic' sensory dendrites. We expressed *Nlg1^untagged* in v'ada neurons and then visualised the anatomy of the motoneurons. In controls, branch growth was restricted exclusively to the inner surface of the muscle (*Figure 8I and J*). This is highlighted by a transverse projection, which shows a relatively uniform layer of motoneuron terminals (*Figure 8K*). In contrast, when v'ada was made to misexpress *Nlg1^untagged*, PM-Mn axon branches grow perpendicular to the normal arborisation between muscle fibres to contact v'ada sensory neurites (*Figure 8L–N*). In several cases further branching from these contacts increased the complexity of these ectopic PM-Mn branches.

One postulate of Vaughn's tropic model of arborisation growth was that the stabilisation of contacts between putative synaptic partners would ultimately lead to functional connectivity (*Vaughn, 1989*). For this to be satisfied, branches generated by such a mechanism should harbour mature synaptic terminals later in development. We looked at axon branches in contact with the dendrites of sensory neurons expressing *Nlg1^untagged* in abdominal fillets stained for the active zone marker BRP (nc82 antibody), and with the neural marker anti-HRP. As shown in *Figure 8O*,

conspicuous concentrations of BRP were found at each contact. Alongside this, we also found strong VGlut immunoreactivity at these ectopic synaptic contacts (data not shown).

## Physiological consequences of Nlg1 disruption

We have shown a very early requirement for Nlg1 for establishing branch structure and that ectopic contacts made by PM-Mns make on v'ada neurites appear to mature into organised hemisynapses. These data show that following contact and stabilisation there is a hierarchy of events that ultimately leads to synapse formation. To explore if Nlg1-Nrx disruptions impact synapse formation we stained against BRP to assess the density of active zones within the branches of *Nlg1* nulls, *Nrx* nulls and animals expressing *Nlg^untagged^* in muscles at pupal stages close to eclosion (*Figure 9A*). Boutons in all these animals were considered as engorgements containing concentrations of synaptic puncta. We found that the density of synapses was not significantly different between *Nlg1* or *Nrx* nulls and wild-type controls, although it was slightly higher in *Nlg1* gain-of-function animals (*Figure 9B*). This suggests that global aspects of active zone development may be largely unaffected by loss of *Nlg1* or *Nrx*. One might expect that functional deficits arising from a failure in appropriate Nlg1 signalling could be due either to defective morphology, changes in transmission, or both. To

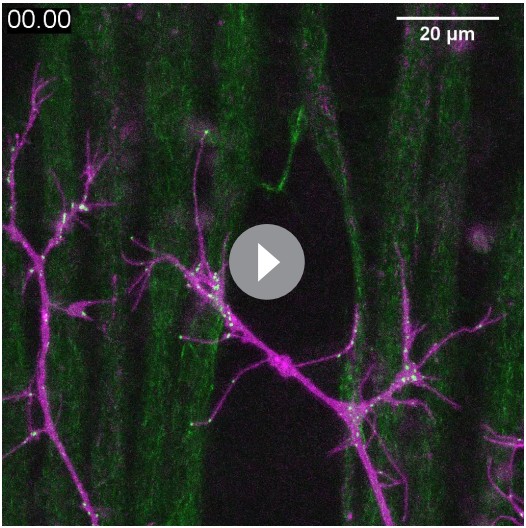

**Video 7.** Nlg1::GFP puncta are correlated with filopodia stability and growth. *Nlg1::GFP* (green) in muscles forms puncta which decorate the axon terminals (magenta), particularly at sites of filopodia growth and at the tips of filopodia (38 hr APF; *Mef2-GAL4 > Nlg1::GFP, VGlut-LexA > myr::tdTomato*). Filopodia which are tipped with stable Nlg1::GFP puncta are significantly longer lived and less motile than those which are not. Filopodia retract only to the point of the last stable Nlg1::GFP punctum. Frames taken at 1 min intervals. Time format: hh:mm.
DOI: https://doi.org/10.7554/eLife.31659.014

explore this, we looked at *GCaMP6m* expressed in the muscles as a proxy for measuring muscle depolarisation at late pupal/pharate stages just prior to eclosion. In wild-type controls, muscle calcium events at this stage usually occur in synchrony across the muscle field (*Figure 9C and D* and *Video 10*). In contrast, in *Nlg1* gain-of-function pupae, calcium events were more common in some groups of fibres than others, resulting in far less synchronicity (*Figure 9E and F* and *Video 11*).

To determine the functional consequence of disruption to Nlg1-Nrx signalling, we also assessed locomotor ability of *Nlg1* null flies in a climbing assay (*Figure 9G*). Using videography and an automated tracking software we found that the climbing speed of *Nlg1* nulls (n = 11) was significantly lower than controls (n = 10) (t(8.15) = 19, p<0.0001, t-test, two-tailed) (*Figure 9H*). We found disruptions to leg motoneuron axon terminals in *Nlg1* nulls, suggesting that some of this deficit may be a direct result of abnormal innervation at the neuromuscular junction(*Figure 9I*).

## Discussion

### PM-Mn axonal arborisations use a dynamic 'synaptotropic-like' mode of growth

Building arborisations of the right size and shape is critical for proper neural circuit development and function. Live imaging studies in vertebrate brains show that neuronal growth is highly dynamic and that structures which appear to be *nascent synapses* play a key role in the development of axonal and dendritic arborisations. These nascent contacts are thought to act like bolts during construction, yet a detailed understanding of their molecular composition and assembly is not known.

To explore this biology, we searched the fly for neurons that grow exuberantly, similar to those in the fish and frog visual systems. The embryonic motor system has been an excellent tool for studying

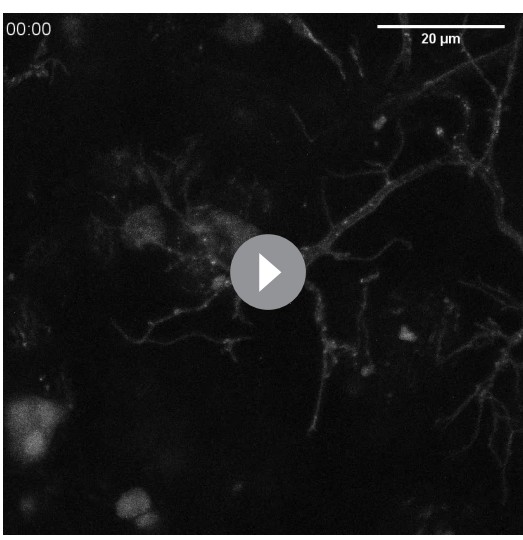

**Video 8.** Nrx::GFP localises to growing axon terminals. *Nrx::GFP* expressed in the PM-Mns (*OK371-GAL4*) faintly coats the axonal membrane, but forms strongly labelled puncta at sites equivalent to Nlg1:: GFP i.e. in branch terminals and at the tips of filopodia (35 hr APF). Frames recorded at 1 min intervals. Time format: hh:mm.
DOI: https://doi.org/10.7554/eLife.31659.015

connectivity and arbor growth, but its small size and rapid development makes it very challenging to image growth live. The postembryonic stages have single identifiable neurons, longer development and more plastic vertebrate-like cell-cell interactions (*Truman, 1990*; *Currie and Bate, 1991*; *Fernandes et al., 1991*; *Consoulas et al., 2000*)

The axonal arborisations of the pleural muscle motoneurons (PM-Mns) are superficial and can be imaged in vivo throughout metamorphosis. Unlike the da sensory neuron input arborisations, which have been useful for studying neurite branch growth (*Williams and Truman, 2004*; *Shimono et al., 2009*; *Yalgin et al., 2015*), PM-Mn axons form synapses; a trait in common with the majority of neuronal arborisations.

PM-Mn growth is very dynamic with a high turnover of exploratory filopodia, with only a few ultimately becoming branches. Their axonal arborisations develop in close association with the pleural muscles and we find that axonal filopodia are stabilised upon contact with myoblasts or immature myotubes (see *Video 2*). A number of in vitro and in vivo studies have previously highlighted that filopodial stability is conferred by the contacts made between potential synaptic partners (*Cooper and Smith, 1992*; *Ziv and Smith, 1996*; *Jontes et al., 2000*).

To determine whether PM-Mn axonal arborisations could be growing using nascent synapses we imaged the active zone marker Bruchpilot (BRP) (*Chen et al., 2014*; *Urwyler et al., 2015*) and vesicular machinery (VGlut and Syt1) and found similar localisations of synaptic proteins, as well as relationships with branch lifetimes as have been described in zebrafish and *Xenopus* (*Alsina et al., 2001*; *Niell et al., 2004*; *Meyer and Smith, 2006*; *Ruthazer et al., 2006*). BRP puncta appear to chart the progression of branch stabilisation events. To our surprise we also found Bruchpilot at similar localisations in the branches of developing eve+ interneurons in the CNS, indicating that this type of growth may not be restricted to the peripheral nervous system but could be common within the fly central nervous system.

## Growing arborisation do not require functional synapses

The exact contributions of synaptogenesis, neural activity and synaptic transmission to the formation of neural networks are unclear. Indeed, there is a rich history regarding the question of whether nervous systems develop in '*forward reference*' to, but without benefit from, functional activity (*Harrison, 1904*; *Weiss, 1941*); see *Haverkamp (1986)* for review.

Interestingly, we find that a great deal of PM-Mn outgrowth takes place prior to robust presynaptic calcium transients (action potentials, <42 hr APF). Furthermore, we found that not until 60 hr APF could we evoke activity in muscles by stimulating the motoneurons. Finally, neither TTX or removing all vesicular neurotransmission, by making *VGlut* null clones, had a significant impact on the mature arbor morphology. Cline and colleagues showed a correlation between the stabilisation of growing retinal ganglion cell axon branches bearing synaptic puncta and visual activity, pointing to use-testing of nascent contacts by activity in a synaptotropic-like mode of growth (*Ruthazer et al., 2006*). In contrast, a number of studies have shown that blocking activity has little impact on morphology, or plays only a role in the refinement of arbor growth (*Haverkamp, 1986*; *Verhage et al., 2000*; *Varoqueaux et al., 2002*; *Hua et al., 2005*; *Ben Fredj et al., 2010*). It appears that different systems vary in their requirement for activity during arbor development.

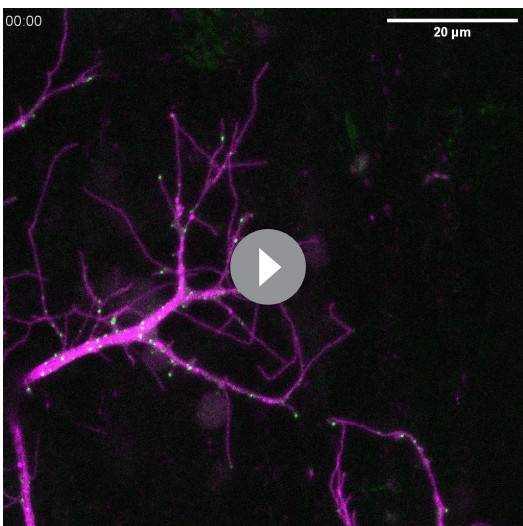

**Video 9.** Liprin-α::GFP puncta are correlated with filopodia stability and growth. *Liprin-α::GFP* (green) expressed in motoneurons (magenta) has a punctate localisation like Nlg1 and Nrx at 35 hr APF. Like Nlg1:: GFP, Liprin-α::GFP puncta at filopodial tips are strongly correlated with filopodial stability and regularly mark the limits of filopodial retraction. (*OK371-GAL4 > mCD8::Cherry + Liprin-α::GFP*). Frames recorded at 30 s intervals. Time format: mm:ss.
DOI: https://doi.org/10.7554/eLife.31659.017

The detailed imaging possible with our system reveals that BRP puncta very rarely enter filopodia and are never stabilised within them. We found the same to be the case for the synaptic vesicle associated proteins Syt1 and VGlut. Since BRP is an important structural component of the cytomatrix at the active zone, these data seemed at odds with the notion that synapses drive branch stabilisation and led us to question if the puncta previously reported at branch nodes in growing retinal ganglion cells really represent bona fide synapses. Importantly, we found that it was not until late stages of development that the size and distribution of BRP puncta became similar to synapses at the larval NMJ (*Fouquet et al., 2009*). In parallel, clusters of glutamate receptors only became conspicuous at late stages, long after arbor shape has been established. Taken together these data suggest that it is only at the later stages of arbor development that synaptic elements pair to form bona fide synapses. The accumulations of presynaptic proteins seen in early arborisations (i.e. 30–40 hr APF) may instead mark stable sites or 'transport hubs' that help store/sort synaptic proteins for a later role in synapse formation.

## Neuritic adhesion complexes (NACs) drive a tropic mode of arborisation growth

If synaptic transmission does not play a role in this type of dynamic arbor growth, what mechanisms are responsible? We find a role for a class of proteins (synaptic cell adhesion molecules), pre-synaptic Neurexins (Nrxs), and their postsynaptic binding partners, Neuroligins (Nlgs).

The *Nlg1* mutants and the clonal FlpStop data suggest that Nlg1 in the muscle is crucial for normal growth. Clonal analysis reveals that branches growing onto muscles with lower levels of *Nlg1* expression are phenotypically similar to those in *Nlg1* nulls. The growth of branches onto territories with more wild-type levels of *Nlg1* expression is reminiscent of the tropic aspect suggested in Vaughn's original hypothesis.

Work in *Xenopus* has previously implicated Nrx-Nlg interactions in arbor growth and posited that synapse formation and subsequent levels of synaptic transmission translates directly into branch stability (*Chen et al., 2010*). Here we propose that these molecules provide adhesion during elaborative phases independent of synapse formation. We describe for the first time that Nlg1::GFP puncta emerge on muscle membranes in vivo following contact with presynaptic filopodia. Nlg1 localises onto the tips and shafts of filopodia and regulates stability. Filopodia that 'capture' such puncta are significantly longer lived than those that do not. Furthermore, time-lapse imaging revealed that Nlg1::GFP act like anchor points by marking the limits of filopodial retraction. This is very different to the growth of zebrafish and mouse neuromuscular junctions, where motoneurons grow between pre-patterned plaques of acetylcholine receptors, as if hopping between stepping stones (*Yang et al., 2001*; *Panzer et al., 2006*; *Jing et al., 2009*).

Presynaptically, we find Nrx puncta on the tips and along the shafts of filopodia i.e. at sites that mirror those of Nlg1. Additionally, we find two other proteins at such sites, Liprin-α and Syd-1, which are known to complex with Nrx. Liprin-α and Syd-1 are found on the membrane and coalesce into dynamic puncta that localise to filopodia tips. Liprin-α puncta appear to limit filopodial retraction, indicating that sites marked by this protein represent the presynaptic counterparts to the structural anchor points marked by Nlg1. The lack of Liprin-α in the long and labile branches of *Nlg1* nulls indicates that a reduction in adhesive strength initially allows unrestrained branch growth, but ultimately

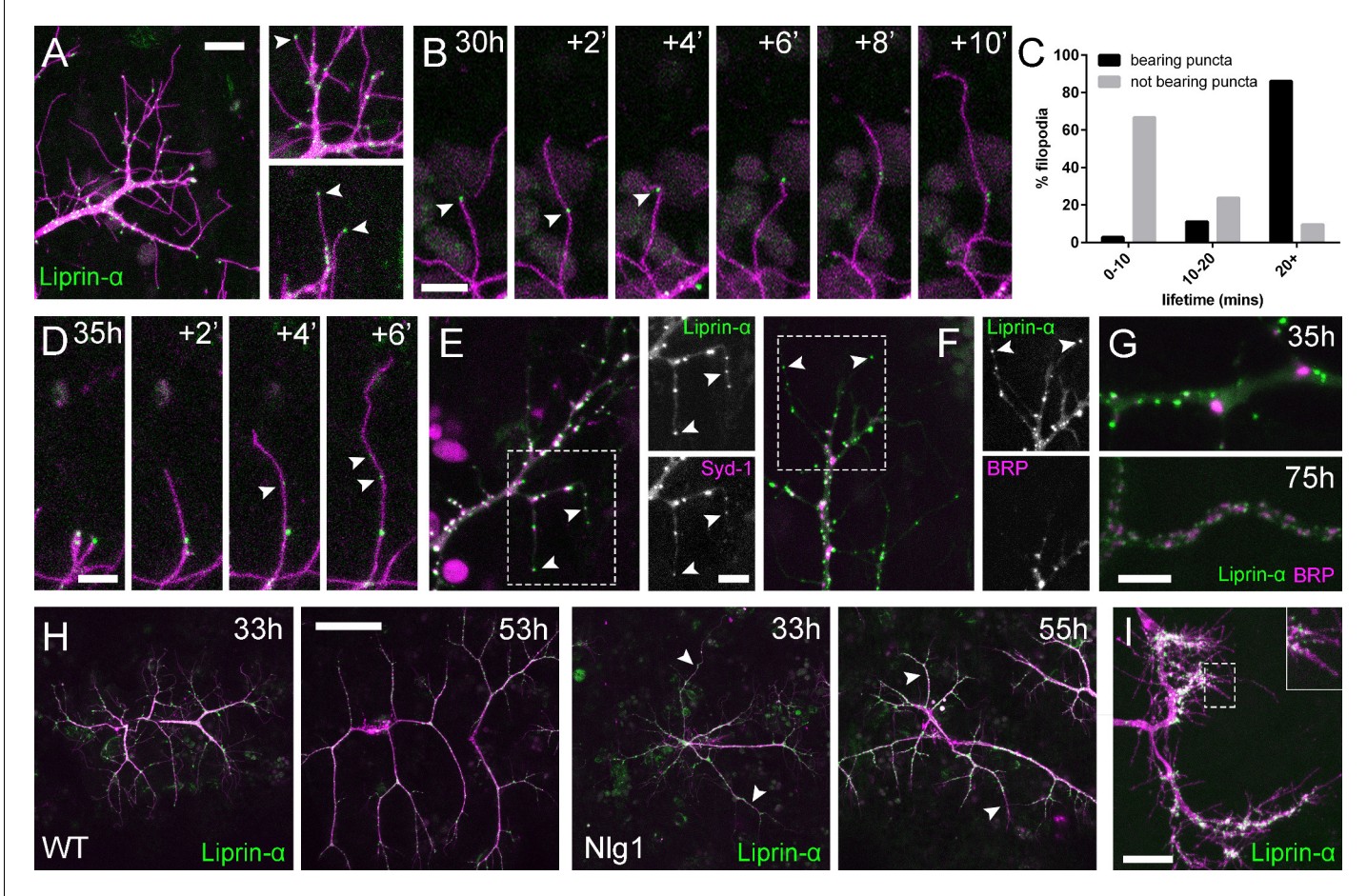

**Figure 7.** Neuritic Adhesion Complexes (NACs) contain Liprin-α and Syd1. (A) Localisation of *Liprin-α::GFP* in growing PM-Mn axonal arborisation at 35 hr APF. *Liprin-α::GFP* (green) expressed with *mCD8::ChRFP* (*OK371-GAL4*; magenta). Cutaway images show Liprin-α::GFP puncta in branches and at tips of filopodia (arrowheads). (B) Detail showing the extension of a filopodium and its retraction only as far as a Liprin-α::GFP punctum (arrowheads). (C) Histogram showing lifetimes of filopodia with Liprin-α::GFP puncta at their tips (n = 36) and without puncta at their tips (n = 21). Filopodia bearing Liprin-α::GFP puncta lived significantly longer than those without puncta (with puncta: 19.44 ± 1.96 mins, without puncta: 8.81 ± 5.88 mins, Mann-Whitney U = 51.5, p<0.0001, two-tailed). (D) Time series shows the rapid precipitation of new Liprin-α::GFP puncta (indicated by arrowheads) within a filopodium. (E) *Liprin-α::GFP* and *Syd-1::Straw* puncta co-localise in growing PM-Mn terminals, including at filopodia tips, (arrowheads). Slight misalignment due to sequentially imaging channels. (F) Liprin-α::GFP and BRP::RFP only co-localise at a few sites. BRP::RFP puncta are not found within filopodia. (G) Liprin::GFP and BRP::RFP puncta occupy distinct subcellular regions at 35hAPF. At 75hAPF Liprin::GFP becomes localised to the edges of BRP::RFP puncta along branches. (H) wild-type and *Nlg1* null arborisations expressing *Liprin-α::GFP* and *mCD8::ChRFP* imaged at 33 hr APF and ~20 hr later. Unusually long, unbranched branches of *Nlg1* nulls (white arrowheads) lack *Liprin-α::GFP* puncta and have collapsed by later stages (I) In *Eve* + ve interneurons *Liprin-α::GFP* forms distinct puncta on the filopodia of growing axon terminals at 24 hr APF. Bars represent SDs. Scale bars: 10 µm (A), 5 µm (B,D,E/F,G), 50 µm (H), 20 µm (I).

DOI: https://doi.org/10.7554/eLife.31659.016

generates unstable branches, which cannot be maintained. This speaks to a balance between adhesion and growth, which is supported by the compacted Nlg1 gain-of-function phenotype.

Previously, Syd-1 and Liprin-α have been found to play a key role in orchestrating synapse assembly by recruiting and retaining other synaptic proteins. Liprin-α, Syd-1 and Nrx puncta at these stages of exuberant growth are transient and do not appear to become 'future synapses'. Their fluidity speaks to a role in branch morphogenesis rather than in a stepwise, clockwork assembly of synaptic machinery at a particular place.

Based on our observations, we propose that these dynamic complexes are composed of a subset of proteins that have been previously called 'synaptic cell adhesion proteins' or drivers of synapse formation. As a placeholder we suggest calling these puncta *neuritic adhesion complexes* or NACs

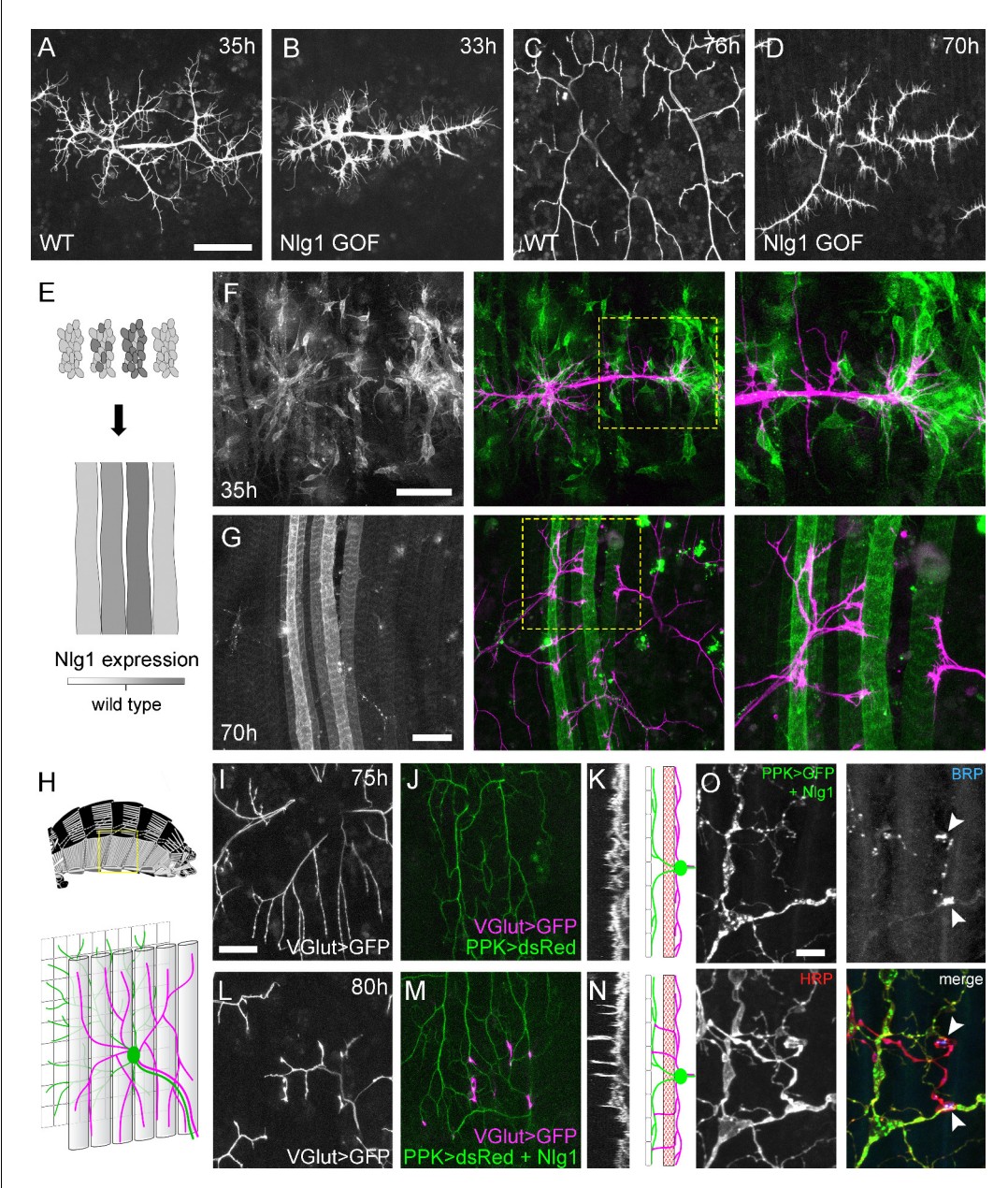

**Figure 8.** *Neuroligin 1* acts locally to drive early arbor growth. (**A–D**) Postsynaptic *Nlg1* overexpression disrupts PM-Mn axon arbor growth. (**A**) Control PM-Mn arborisations at 35 hr APF and (**B**), PM-MN Arborisations at 33 hr APF in a background of postsynaptic *Nlg1^untagged* expression (*Mef2-GAL4; VGlut-LexA > myr::GFP*). (**C**) Control arborisations at 76 hr APF. (**D**) arborisations in a background of muscle specific *Nlg1^untagged* expression at 70 hr APF (**E**) Schematic of 'flip-out' clones generation. Muscle precursor clones induced with *hsFlp* at larval L3 stages produce clusters of GAL80 negative myoblasts which fuse to form 'stripes' of clonal muscle fibres expressing *Nlg1^untagged* and *myr::tdTomato*. (**F**) At 35 hr APF, PM-Mn motoneuron branches (magenta) show preferential elaboration onto clonal *Nlg1^untagged* expressing myotubes and myoblast clusters (green). (**G**) Axonal branches in contact with clonal, *Nlg1^untagged* expressing muscle fibres show a hyper-stabilisation phenotype. Growth of branches appears to be preferentially directed along clonal fibres. (**H–O**) Ectopic expression of Nlg1 in class IV da sensory neurons drives changes in motoneuron axonal arbor morphology. (**H**) Schematic shows the relative positions of class IV v'ada sensory input arborisations (green), pleural muscles (grey tubes) and motor axon arborisations (magenta) in the pleural abdominal body wall (schematic of musculature adapted from *Demerec, 1950*). (**I**) Motoneuron axon terminals expressing *myr::GFP* (*VGlut-LexA*) at 80 hr APF in a control (**J**) the v'ada input arborisations in the same region expressing *dsRed* (*PPK-Gal4; Grueber et al., 2003*). (**K**) Transverse projection shows the PM-Mn axon arborisations restricted to a single plane. Schematic transverse shows the separation of the v'ada arborisation (green) from the motor axon terminals (magenta) by the pleural muscles (red). (**L–N**) *Nlg1^untagged* expression in class IV da sensory neurons results in axonal branches that penetrate gaps between the muscle fibres and make contact with the sensory arborisations. These aberrant branches are shown in the transverse view and run perpendicularly to the rest of the arborisation. (**O**) Abdominal fillet of a newly eclosed adult

*Figure 8 continued on next page*

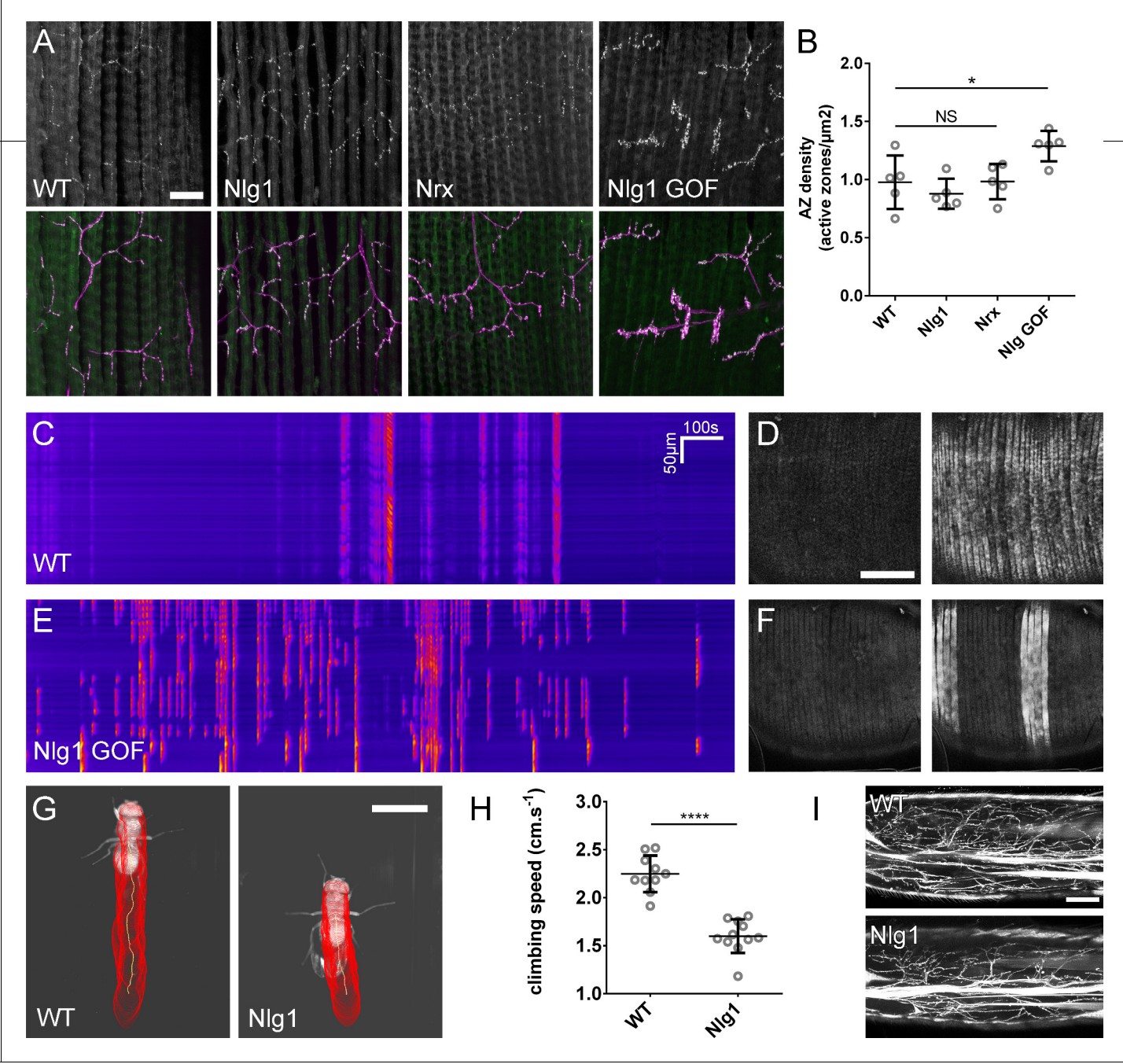

**Figure 9.** The roles of *Nlg* and *Nrx* in NAC dependent arbor construction appear decoupled from synapse formation and neurotransmission. (A–B) Synapse formation at the light microscope level appears largely unaffected by disruptions to Nrx-Nlg signalling. (A) Active zones at the pleural neuromuscular junctions in adult abdominal fillets from a wild-type control, an *Nlg1* null, an *Nrx* null and an *Nlg1* gain-of-function (*Mef2-GAL4 > Nlg1* untagged) revealed by antibody staining against BRP. (B) Active zone densities calculated from BRP puncta in synaptic terminals. No significant differences were found between the densities of active zones in controls (0.98 ± 0.23 $\mu m^{-2}$, n = 5) and *Nlg1* nulls (0.88 ± 0.13 $\mu m^{-2}$, n = 5) or controls and *Nrx* nulls (0.98 ± 0.15 $\mu m^{-2}$, n = 5) (*Nlg1* vs controls: Mann-Whitney U = 9, p=0.53, two-tailed. *Nrx* vs controls: Mann-Whitney U = 12, p=0.94, two-tailed). A small, but significant difference was found between active zone densities in *Nlg1* gain-of-function terminals (1.29 ± 0.13 $\mu m^{-2}$, n = 5) and controls (Mann-Whitney U = 1, p=0.0159, two-tailed). (C–F) *Nlg1* overexpression causes changes in PM-Mn axonal arbor morphology, which result in altered connectivity and transmission. (C) GCaMP6m Δfs in a control and in a pupa expressing *Nlg1^untagged^* in muscles (*Mef2-GAL4*) staged within 10 hr of eclosion, (D). In controls, calcium events are generally synchronous across the pleural muscle field whereas in gain-of-function pupae events are often restricted to smaller muscle subsets. This is shown by breaks in the light-coloured bars on the kymograph and in the sequential images (D and F). (G–I) Loss of *Nlg1* has a pronounced impact on locomotor ability (G) Automated body tracking performed using FlyLimbTracker (*Uhlmann et al., 2017*) in Icy (*de Chaumont et al., 2012*) of a control and an *Nlg1* null, 2 days post-eclosion (H) Control flies have a significantly faster climb speed than *Nlg1*

*Figure 9 continued*

nulls (controls: 2.25 ± 0.19 cm.s$^{-1}$, n = 10. *Nlg1*: 1.60 ± 0.18 cm.s$^{-1}$, n = 11. t(8.15) = 19, p<0.0001, t-test, two-tailed). (I) Motoneuron axon terminals (*VGlut-LexA > myr::GFP*) innervating the femur 1 day post-eclosion of an *Nlg1* null and of a control. Bars represent SDs. Scale bars: 20 µm (**A**), 100 µm (**D,F,I**), 0.2 cm (**G**).

DOI: https://doi.org/10.7554/eLife.31659.019

to highlight their role in axon growth *prior to synapse formation*. That we see direct evidence for the formation of these NACs on filopodia themselves suggests that dynamic interactions between filopodia and the postsynaptic target are fundamental determinants of tree construction.

This early role for Nlg1/Nrx based NACs in arbor growth is supported by strong evidence that exploratory filopodia and branch tips are highly sensitive to changes in the early expression of postsynaptic *Nlg1*. Our gain-of-function myoblast/myotube clones seem to have a 'hyper-stabilising' effect on wild-type neurons and ultimately constrain growth. Moreover there is some indication of preferential growth into regions with elevated postsynaptic Nlg1 expression. We were very surprised to see wildtype PM-Mns grow past muscles, into novel territories and make contacts with class IV da sensory neurons ectopically expressing Nlg1.

## Physiological consequences of Nlg1 disruption

The precise role that Nrxs and Nlgs play in nervous system development has been broadly disputed. Previous work has shown the strong synaptogenic potential of these proteins by expressing them in HEK cells or on micropatterned substrates, revealing them to be potent regulators of hemisynapse formation (*Scheiffele et al., 2000*; *Dean et al., 2003*; *Graf et al., 2004*; *Chih et al., 2005*; *Lee et al., 2010*; *Czöndör et al., 2013*). Others have shown that mice lacking three Nlg homologues or all three α-Nrxs build brains with grossly normal cytoarchitectures and synapse densities (*Missler et al., 2003*; *Varoqueaux et al., 2006*) and posit that their role is solely to modulate synapse function. The elegant studies of *Kwon et al. (2012)* suggest a cell autonomous role for Nlg in vertebrate synapse formation similar to our clonal studies here. Roles for Nlg and Nrx in synapse formation have previously been described in the fly (*Li et al., 2007*; *Banovic et al., 2010*; *Chen et al., 2012*).

Teasing apart the function of such gene families in vertebrates can been difficult due to the multiple copies of these genes and their degeneracy with other proteins (*Sugita et al., 2001*; *Ko et al., 2009*; *Siddiqui et al., 2010*; *Xu et al., 2010*; *Ko et al., 2011*; *Soler-Llavina et al., 2011*). Here in this simpler system we show that Nlg1/Nrx based adhesion complexes play a key role in the early growth of axonal arborisations prior to the formation of functional synapses. In addition, we find that disrupting Nlg1/Nrx signalling ultimately has little effect on the timing of synapse development and size of active zones, suggesting that tree morphogenesis is uncoupled from the formation of synapses. One prediction from this is that disruptions in functional connectivity could largely be due to disruptions in growth and morphology of the arborisation. An important observation that frames this is the outcome of ectopically expressing Nlg1 in class IV da sensory neurons. There we see the generation of a 'synthetic connectivity' on the normally 'asynaptic' da sensory neurites, with the development of mature, differentiated presynaptic structures in later pupal stages i.e. sites marked by BRP and synaptic vesicle proteins.

In support of this, our calcium imaging reveals that the normal patterns of muscle activity are disrupted when Nlg1 is overexpressed, with changes in the frequency and spatial heterogeneity of calcium events. Since elevated postsynaptic Nlg1 expression results in condensed arborisations, we propose that this is a result of irregular innervation. The behavioural analysis shows that Nlg1 null adults had significant motor behaviour deficits. Taken together these data provide a link between an adhesion-based mechanism of arbor growth and synaptic connectivity, which ultimately affects function.

## Conclusions

In this study we have demonstrated, for the first time outside of vertebrates, a dynamic 'synaptotropic-like' mode of arbor growth. In contrast to the mechanisms that have previously been proposed, we find no evidence that functional synapses drive this type of growth. Instead, our data

point to dynamic NACs, composed of Neurexin, Neuroligin, Liprin-α and Syd-1, playing a role by stabilising exploratory filopodia and branches. Alongside this, our evidence showing similar subcellular localisations for these proteins in *Drosophila* central neurons points toward this mode growth being a universal mechanism for building complex trees. It may be that Berry and colleagues' emphasis on filopodia in their 'synaptogenic filopodial theory' was correct. It is an appealing idea that such mechanisms could construct axonal and dendritic arborisations of a broad spectrum of shapes and sizes by making only subtle changes to a 'stick and grow' algorithm. Exactly which proteins constitute these adhesion complexes and how they interact with the machinery regulating cytoskeletal remodelling will be a focus of future investigation.

# Materials and methods

**Key resources table**

| Reagent type (species) or resource | Designation | Source or reference | Identifiers | Additional information |
|---|---|---|---|---|
| genetic reagent (*D. melanogaster*) | OK371-GAL4 | Bloomington Drosophila Stock Centre | BDSC:26160 | |
| genetic reagent (*D. melanogaster*) | Mef2-GAL4 | Bloomington Drosophila Stock Centre | BDSC:27390 | |
| genetic reagent (*D. melanogaster*) | PPK-GAL4 | Bloomington Drosophila Stock Centre | BDSC:32078 | |
| genetic reagent (*D. melanogaster*) | VGlut$^{NMJX}$-GAL4 | *Daniels et al. (2008)* | | P{VGlut-GAL4.D}NMJX |
| genetic reagent (*D. melanogaster*) | VGlut(Trojan)-T2A-LexA | Bloomington Drosophila Stock Centre | BDSC:60314 | |
| genetic reagent (*D. melanogaster*) | Df(2L)vglut$^2$ | *Daniels et al. (2006)* | | |
| genetic reagent (*D. melanogaster*) | Nlg1$^{I960}$ | *Banovic et al. (2010)* | | |
| genetic reagent (*D. melanogaster*) | Nlg1$^{ex2.3}$ | *Banovic et al. (2010)* | | |
| genetic reagent (*D. melanogaster*) | Nrx$^{241}$ | *Li et al. (2007)* | | |
| genetic reagent (*D. melanogaster*) | Df(3R)BSC747 | Gift from Hermann Aberle | FlyBase: FBab0045813 | |
| genetic reagent (*D. melanogaster*) | Df(3R)Exel6191 | Gift from Hermann Aberle | FlyBase: FBab0038246 | |
| genetic reagent (*D. melanogaster*) | RN2-Flp | *Roy et al. (2007)* | | |
| genetic reagent (*D. melanogaster*) | UAS-myr::GFP | *Pfeiffer et al., 2012* | | |
| genetic reagent (*D. melanogaster*) | UAS-cytoplasmicGFP | *Pfeiffer et al., 2012* | | |
| genetic reagent (*D. melanogaster*) | UAS-Lifeact::Ruby | Bloomington Drosophila Stock Centre | BDSC:35545 | |
| genetic reagent (*D. melanogaster*) | UAS-CLIP170::GFP | *Stramer et al., 2010* | | |
| genetic reagent (*D. melanogaster*) | UAS-mCD8::ChRFP | Bloomington Drosophila Stock Centre | BDSC:27391 | |
| genetic reagent (*D. melanogaster*) | UAS-myr::tdTomato | Bloomington Drosophila Stock Centre | BDSC:30124 | |
| genetic reagent (*D. melanogaster*) | UAS-BRP::RFP | Gift from Stephan Sigrist | | |
| genetic reagent (*D. melanogaster*) | UAS-Nlg1::GFP | *Banovic et al. (2010)* | | |
| genetic reagent (*D. melanogaster*) | UAS-Nlg1untagged | *Banovic et al. (2010)* | | |
| genetic reagent (*D. melanogaster*) | UAS-Nrx::GFP | *Banovic et al. (2010)* | | |
| genetic reagent (*D. melanogaster*) | UAS-Liprin-α::GFP | *Fouquet et al. (2009)* | | |
| genetic reagent (*D. melanogaster*) | UAS-Syd1::Straw | *Owald et al. (2010)* | | |
| genetic reagent (*D. melanogaster*) | UAS-GCaMP6M | Bloomington Drosophila Stock Centre | BDSC: 42750 | |
| genetic reagent (*D. melanogaster*) | UAS-GCaMP6M | Bloomington Drosophila Stock Centre | BDSC: 42748 | |
| genetic reagent (*D. melanogaster*) | GluRIIE::GFP | FlyFos TransgeneOme (fTRG) library | fTRG:154 | |
| genetic reagent (*D. melanogaster*) | BRP::GFP | Bloomington Drosophila Stock Centre | BDSC:59411 | |
| genetic reagent (*D. melanogaster*) | VGlut::GFP | Bloomington Drosophila Stock Centre | BDSC:59292 | |
| genetic reagent (*D. melanogaster*) | Syt1::GFP | Bloomington Drosophila Stock Centre | BDSC:59788 | |
| genetic reagent (*D. melanogaster*) | Nlg1$^{FlpStop}$ | FlpStop construct from *Fisher et al. (2017)* | | |
| genetic reagent (*D. melanogaster*) | LexAop-myrGFP | Bloomington Drosophila Stock Centre | BDSC:32210 | |

*Continued on next page*

*Continued*

| Reagent type (species) or resource | Designation | Source or reference | Identifiers | Additional information |
|---|---|---|---|---|
| antibody | antiVglut (C-term) (rabbit polyclonal) | *Mahr and Aberle (2006)* | RRID:AB_2490071; aa 561–632 | (1:10,000) |
| antibody | antiBRP (nc82) (mouse monoclonal) | Developmental Studies Hybridoma Bank | RRID:AB_231486; aa 1227–1740 | (1:5) |
| antibody | antiGFP (chicken polyclonal) | Thermo Fisher Scientific (Invitrogen) | Cat# PA1-86341, RRID:AB_931091 | (1:1000) |
| antibody | antiHRP-Cy3 (goat polyclonal) | Jackson Immunoresearch | | (1:5000) |
| antibody | Alexa-488 secondary | Thermo Fisher Scientific (Invitrogen) | | (1:500) |
| antibody | Cy3 and Cy5 secondaries | Jackson Immunoresearch | | (1:500) |

## Fly stocks

Flies were reared on a standard yeast-cornmeal-molasses diet. For time-sensitive experiments flies were raised at 25°C, or at room temperature if precise staging was not required.

OK371-GAL4; UAS-myr::GFP

OK371-GAL4; UAS-myr::GFP, UAS-BRP::RFP

OK371-GAL4; UAS-Lifeact::Ruby/UAS-CLIP170::GFP

VGlut(Trojan)-T2A-LexA, LexAop-myr::GFP

VGlut(Trojan)-T2A-LexA, LexAop-myr::GFP; Mef2-GAL4, UAS-mCD8::ChRFP

VGlut(Trojan)-T2A-LexA, LexAop-myr::GFP; Mef2-GAL4, UAS-myr::tdTomato

OK371-GAL4, UAS-mCD8::ChRFP/BRP::GFP

OK371-GAL4, UAS-mCD8:ChRFP/VGlut::GFP

OK371-GAL4, UAS-mCD8:ChRFP/Syt1::GFP

RN2-Flp, Tub-FRT-CD2-FRTGal4, UAS-mCD8GFP/UAS-BRP::RFP

OK371-GAL4, UAS-mCD8:ChRFP/GluRIIE::GFP

OK371-GAL4, UAS-GCaMP6m

VGlut(Trojan)-T2A-LexA; Mef2-GAL4, UAS-GCaMP6m/LexAop-TRPA1

$VGlut^{NMJX}$-Gal4, UAS-mCD8::GFP, $hsFlp^{122}$; $FRT^{40A}$, $Df(2L)VGlut^2$; UAS-myr::GFP

$VGlut^{NMJX}$-Gal4, UAS-mCD8::GFP, $hsFlp^{122}$; $FRT^{40A}$, $Df(2L)VGlut^2$/$FRT^{40A}$; UAS-myr::GFP

Elav-GAL4, $hsFlp^{122}$; $FRT^{40A}$, $Df(2L)vglut^2$/$FRT^{40A}$; UAS-cytoplasmicGFP

VGlut(Trojan)-T2A-LexA, LexAop-myr::tdTomato; Mef2-GAL4, UAS-Nlg1::GFP

UAS-Nrx::GFP; OK371-GAL4, UAS-mCD8::ChRFP

OK371-GAL4, UAS-mCD8::ChRFP/UAS-Liprin-$\alpha$::GFP

OK371-GAL4, UAS-Liprin-$\alpha$::GFP/UAS-Syd1::Straw

OK371-GAL4, UAS-Liprin-$\alpha$::GFP; UAS-BRP::RFP

OK371-GAL4, UAS-Liprin-$\alpha$::GFP; UAS-BRP::RFP; $DF(3R)BSC747$/$Nlg1^{ex2.3}$

RN2-Flp, Tub-FRT-CD2-FRT-GAL4, UAS-mCD8::mCD8::ChRFP/UAS-Liprin-$\alpha$::GFP

VGlut(Trojan)-T2A-LexA, LexAop-myr::GFP; $Df(3R)BSC747$/$Nlg1^{ex2.3}$

VGlut(Trojan)-T2A-LexA, LexAop-myr::GFP; $Df(3R)BSC747$/$Nlg1^{I960}$

VGlut(Trojan)-T2A-LexA, LexAop-myr::GFP; $Df(3R)Exel6191$/$Nrx^{241}$

VGlut(Trojan)-T2A-LexA, LexAop-myr::GFP; Mef2-GAL4/UAS-$Nlg1^{untagged}$ $hsFlp^{122}$; VGlut(Trojan)-T2A-LexA, LexAop-myr::GFP, $\alpha$Tub84B-FRT.GAL80; Mef2-GAL4, UAS-myr::tdTomato/UAS-$Nlg1^{untagged}$

PPK-GAL4, UAS-dsRed/VGlut LexA, LexAop-myr::GFP

PPK-GAL4, UAS-dsRed/VGlut LexA, LexAop-myr::GFP; UAS-$Nlg1^{Untagged}$

PPK-GAL4, UAS-mCD8::GFP; UAS-$Nlg1^{untagged}$

VGlut(Trojan)-T2A-LexA, LexAop-myr::GFP; $Nlg1^{FlpStop\ D}$/$Df(3R)BSC747$

VGlut(Trojan)-T2A-LexA, LexAop-myr::GFP; $Nlg1^{FlpStop\ ND}$/$Df(3R)BSC747$

$hsFlp^{122}$; VGlut(Trojan)-T2A-LexA, LexAop-myr::GFP/Mef2 GAL4; $Nlg1^{FlpStop\ ND}$/$Df(3R)BSC747$

$hsFlp^{122}$; VGlut(Trojan)-T2A-LexA, LexAop-myr::GFP/Mef2 GAL4; $Nlg1^{FlpStop\ D}$/$Df(3R)BSC747$

Mef2-Gal4, UAS-GCaMP6m; UAS-$Nlg1^{Untagged}$

Mef2-Gal4, UAS-GCaMP6m

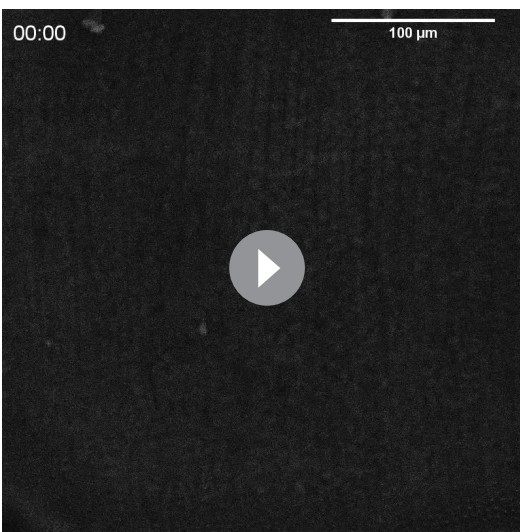

00:00                                    100 μm

**Video 10.** Functional consequence of disrupted Nlg1 signalling; control. GCaMP6m was used as a proxy to explore the effect of disrupted Nlg1 signalling on late-stage postsynaptic activity. In controls, calcium activity within 10 hr prior to eclosion is characterised by changes in fluorescence which are synchronous across the muscle field (*Mef2-GAL4 > GCaMP6m*). Frames recorded at 1 s intervals. Time format mm:ss.
DOI: https://doi.org/10.7554/eLife.31659.020

## Imaging

Imaging was performed at room temperature using Zeiss LSM 510 or 800 series confocal microscopes and EC Plan-Neofluar 20x/0.50 or Plan-Apochromat 40x/1.3 objectives.

## Mounting and live imaging

For consistent developmental staging, pupae were collected at 0 hr APF (white prepupal stage) and incubated on moist tissue paper at 25°C in parafilm sealed petri dishes. In preparation for mounting, pupae were dried on tissue paper, immobilised on strips of double sided sticky tape and carefully removed from their puparial cases using forceps. Pupae were mounted using 22 × 22 mm cover slips beneath a thin coating of halocarbon oil. For imaging sessions of greater than 1 hr pupae were mounted in humidity maintained chambers consisting of a platform of semi-permeable membrane suspended above a hole in a specialised steel slide and sealed with a watertight ring of petroleum jelly.

For experiments requiring temperature manipulation, a specialised temperature adjustable stage was built consisting of a glass slide fastened to a 5 × 5 cm thermoelectric Peltier device (Maplin Electronics Ltd., UK) which was mounted on a regular microscope slide-holder. Before each experiment the temperature of the stage was calibrated using a thermocouple probe attached to a multi-meter device (Rapid Electronics Ltd., UK) and controlled from a 0–30 v power supply unit.

## Mosaic analysis

The MARCM method was used to generate and label *VGlut* null motoneuron clones. A preliminary MARCM screen using OK371-GAL4 found that the pleural muscle motoneurons are born during the embryonic wave of neurogenesis. To generate clones during this wave of neurogenesis, breeding adults were allowed to lay on grape jelly plates with yeast for 2 hr at 25°C. Eggs were incubated for 3 hr at 25°C and heat shocked by incubation in a water bath at 37°C for 45 min, followed by resting at room temperature for 30 min, followed by a further 30 min at 37°C. Upon hatching, larvae were transferred to standard food and raised at 25°C.

To generate small numbers of GAL80 'flip-out' or FlpStop Mef2-GAL4 positive muscle clones L3 wandering larvae were heat shocked for 20 min by incubation in a water bath at 37°C.

## Dissections and immunocytochemistry

Dissections were made in Sylgard silicone elastomer (Dow Corning, USA) lined petri dishes in 1x phosphate buffered saline (PBS; pH 7.3, Thermo Fisher Scientific, USA).

For dissections of pupal abdominal body walls, pupae were removed from their puparial cases and pinned via the head using electrolytically sharpened 0.1 mm tungsten pins. The posterior tip of each abdomen was removed (from approximately segment 6) and an incision made along the dorsal midline until reaching the thorax. Abdomens were separated from the thorax by cutting along the joint and laid flat using additional tungsten pins. Viscera were removed by a combination of gentle pipetting and forceps. Larval body wall dissections were performed as described previously (*Broadie and Bate, 1993*)

Dissected samples were fixed in buffered 3.6% formaldehyde for 45 min at room temperature and washed three times in PBS containing 0.3 Triton-X100 (Sigma, USA) (PBST). Following fixation, samples were blocked in PBST containing 4% goat serum (Sigma, USA) for 45 min and incubated in

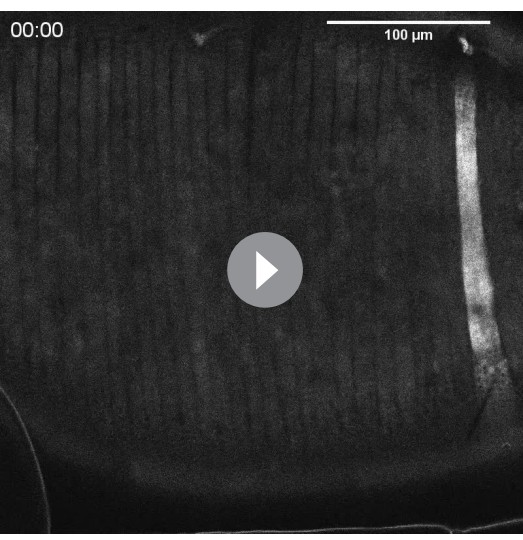

**Video 11.** Functional consequence of disrupted Nlg1 signalling; Nlg1 GOF. GCaMP6m was used as a proxy to explore the effect of disrupted Nlg1 signalling on late-stage postsynaptic activity. *Nlg1* overexpression in muscles results in heterogenous muscle calcium activity, with subsets of muscles firing more frequently than others (*Mef2-GAL4 > GCaMP6m + Nlg1^untagged*). Frames recorded at 1 s intervals. Time format mm:ss.
DOI: https://doi.org/10.7554/eLife.31659.021

solutions of primary antibodies made up with PBST overnight at 5°C. After rinsing three times with PBST over the course of a day, samples were incubated in solutions of secondary antibodies made up with PBST overnight at 4°C. After three rinses with PBST and a final rinse with PBS, samples were mounted on poly-L-lysine coated coverslips, dehydrated through a series of alcohol, cleared with xylene and mounted in DePeX (BDH Chemicals, UK).

Primary antibodies were used at the following concentrations: mouse anti-nc82 (DSHB, USA) 1:5, rabbit anti-VGlut-C term (Gift from Hermann Aberle; *Mahr and Aberle, 2006*) 1:10,000, chicken anti-GFP (Invitrogen, USA) 1:1000. Secondary antibodies were used at the following concentrations: Cy3 goat anti-HRP (Jackson Immunoresearch, USA) 1:5000, AlexaFluor 488 goat anti-Chicken (Invitrogen, USA) 1:500, Cy5 donkey anti-mouse 1:500, Cy5 donkey anti-rabbit 1:500 (Jackson Immunoresearch, USA).

For dissections of CNSs, white pre-pupae were collected and incubated at 25°C. Pupae were dissected from their puparial cases as described above. CNSs were dissected out in 1x PBS in Sylgard lined Petri-dishes. These were then immediately fixed in 3.6% formaldehyde for 20 min. Fixative was washed off thoroughly using 1x PBS. CNSs were mounted on glass slides in Slow-Fade Antifade reagent (Thermo Fisher Scientific) and sealed beneath coverslips using nail-varnish.

## Microinjections

To prepare for injection, pupae were dried on tissue paper and secured to glass slides using double sided sticky tape. Needles, pulled from glass capillaries, were loaded with 0.5 mM TTX made up in PBS or a control solution of PBS. Injections were made into the abdominal cavities through small access holes made in the puparial cases, using a Transjector-5246 micro-injector (Eppendorf, Germany). Following injection, animals were removed from the sticky tape with a damp paintbrush and incubated at 25°C.

## Climbing assay

Adult females were used two days post-eclosion. To assess climb speed, each fly was placed in a square plastic tube with 1 cm edges and transparent walls. The narrow calibre prevented flies from jumping or flying. Each fly was tapped to the bottom and its climb recorded with an S-PRI high-speed camera (AOS Technologies AG, Switzerland) at 700 fps, frame size of 900 × 700 pixels, under red light illumination. Automated body tracking was performed using FlyLimbTracker (*Uhlmann et al., 2017*) in Icy (*de Chaumont et al., 2012*) over a distance of 0.27 to 0.8 cm. Only recordings in which the climbing path had a linearity over 90%, as determined with the Motion Profiler processor in Icy, were considered for analysis. The average climb speed of each fly was determined by calculating the displacement of the centroid of the body between each frame over three repeats.

## Image analysis

Raw image stacks were imported into the image processing software FIJI (http://imagej.net/Fiji) for enhancement of brightness and contrast. For clarity, the freehand select tool was used to remove

obscuring objects created by degenerating larval tissues and macrophages on a slice by slice basis. Collapsed z-projections were imported into Photoshop (Adobe, USA) for figure assembly. Schematic cartoons were made in Photoshop or Illustrator (Adobe, USA). To generate kymographs the re-slice tool in FIJI was used to reconfigure the axis of image sequences.

### Statistical analysis

Statistical analyses were performed using the software package GraphPad Prism (GraphPad Software Inc., USA). Significances correspond to the following p values: $\leq 0.05$ = *, $\leq 0.01$ = **, $\leq 0.001$ = ***, $\leq 0.0001$ = ****. For selecting between parametric and non-parametric tests, the residuals of each data set were tested for normality. Results are reported with standard deviations (SDs).

### Quantification of synaptic puncta

Relationships between BRP::RFP, Nlg1::GFP and Liprin-$\alpha$::GFP puncta dynamics and branch growth were calculated by manually counting and tracking puncta in FIJI. Puncta were identified by accumulations of fluorescent protein measuring three or more pixels in diameter at the native resolution. The diameters of BRP::RFP puncta were measured by intersection with the line tool in FIJI at their widest point and calculation of the full width at half maximum of the peak in fluorescence.

### Morphometric analysis

Arbor reconstructions were generated using the semi-automated plugin for FIJI, Simple Neurite Tracer. Reconstructed skeletons were imported into a bespoke analysis software in which total arbor length, total branch number, Strahler order and total arbor were computed. Arbor area was calculated by giving branch coordinates a hypothetical 'contact distance' which represented its area of coverage. This value was set to 20 µm. In addition, this software could be used to add and delete vertices and nodes, allowing corrections to me made to tracing errors.

### Bendiness index

To calculate branch bendiness, the 'actual' length of primary, secondary and tertiary axon branches was measured using the freehand line tool in FIJI as the distance between their nodes (or in the case of primary branches, between nodes and tips). These values were divided by the direct distances between these nodes, measured using the straight-line tool, giving an index of bendiness.

### Active zone density

Active zone densities in axon terminals were calculated manually in FIJI using the multi-point tool. Bouton areas were calculated by tracing around boutons using the freehand selection tool.

## Acknowledgements

We would like to thank Hermann Aberle for providing the Nlg1 alleles and UAS reagents as well as the OK371-GAL4 enhancer-trap line and anti-VGlut antibodies, Stephan Sigrist for providing the Liprin-$\alpha$ and Syd1 reagents, Barret Pfeiffer for providing the UAS-GFP lines with translational enhancers, Aaron DiAntonio for sharing the *Df(2L)VGlut$^2$* allele, Frank Schnorrer for providing a copy of Mef2-GAL4, K VijayRaghavan for sending the FlyFos GluRIIE::GFP line and Tom Clandinin for kindly sharing his FlpStop construct and for advice on implementing it. In addition, we thank Jon Clarke for generously allowing us to use his confocal microscope and Maria Catalina Moreno Rodriguez for technical support. We also thank Matthias Landgraf for providing the VGlut-LexA, discussions and advice on experiments. Finally, we thank Jon Clarke, Matthias Landgraf, Martin Meyer and Laura Andreae for reading the manuscript. This work was funded by the BBSRC.

# Additional information

## Funding

| Funder | Grant reference number | Author |
|---|---|---|
| Biotechnology and Biological Sciences Research Council | BB/L022672/1 | William D Constance<br>Amrita Mukherjee<br>Yvette E Fisher<br>Sinziana Pop<br>Eric Blanc |

The funders had no role in study design, data collection and interpretation, or the decision to submit the work for publication.

## Author contributions

William D Constance, Conceptualization, Resources, Data curation, Validation, Investigation, Visualization, Methodology, Writing—original draft, Writing—review and editing; Amrita Mukherjee, Sinziana Pop, Investigation, Methodology, Writing—review and editing; Yvette E Fisher, Resources, Writing—review and editing; Eric Blanc, Software, Built the software from scratch after discussions about morphometry and neurite shape, Tested the software and refined the code to fit to purpose; Yusuke Toyama, Conceptualization, Resources, Supervision, Funding acquisition, Visualization, Project administration, Writing—review and editing; Darren W Williams, Conceptualization, Resources, Supervision, Funding acquisition, Investigation, Visualization, Methodology, Writing—original draft, Project administration, Writing—review and editing

## Author ORCIDs

William D Constance ID http://orcid.org/0000-0002-2514-7039
Sinziana Pop ID http://orcid.org/0000-0002-8811-8307
Eric Blanc ID http://orcid.org/0000-0002-4369-0254
Darren W Williams ID http://orcid.org/0000-0001-5917-4935

## Decision letter and Author response

Decision letter https://doi.org/10.7554/eLife.31659.024
Author response https://doi.org/10.7554/eLife.31659.025

# Additional files

## Supplementary files

• Transparent reporting form
DOI: https://doi.org/10.7554/eLife.31659.022

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
