## [Decision Letter]

Thank you for sending your article entitled "Neurexin and Neuroligin-based adhesion complexes drive axonal arborisation growth independent of synaptic activity" for peer review at *eLife*. Your article has been reviewed by two reviewers, and the evaluation has been overseen by K VijayRaghavan as the Senior and Reviewing Editor.

Reviewer #1:

The manuscript by Constance and colleagues describes a novel imaging approach and molecular genetic analysis of the development branched axonal arbours of adult *Drosophila* motor neurons. The manuscript is well written, referenced and experiments clearly described. The manuscript makes two primary claims – that neuronal activity and vesicular neurotransmission have little or no influence on the development of early synaptic contacts in adult neurons and the Neuroligin-Neurexin signalling complex plays a key role. This first conclusion seems very well established by the experiments provided, fitting with an emerging theme of neurotransmission independence for most synapses in both invertebrates and vertebrates (excluding retinal bipolar neurons and a few other exceptions). The second conclusion is also reasonably supported and the authors have dealt well with a drawback of the system in that postsynaptic muscles are polynucleate limiting the effectiveness of clonal techniques. The majority of the manuscript makes use of a live imaging technique to evaluate synapse development. I would like to have seen this technique applied to Nlg and Nrx mutants more extensively to provide clearer insight into exactly which step of synaptogenesis is perturbed by these factors. Overall, even though the ultimate conclusions may not be entirely unexpected, the manuscript provides insight into a novel system to study synapse placement, assembly and development.

Reviewer #1 (Minor Comments):

The authors should carefully consider their use of the term "synaptotropic" to avoid confusion. The term has evolved to be used often to denote synapse growth and maturation rather than the original coinage.

The number of animals and terminals examined should be added to some key observations e.g. Figure 6, Figure 7. as in the other legends.

*Reviewer #2:*

The authors establish a preparation to study in vivo the arborisation of neuronal processes. The preparation they use are the motoneurons that innervate the abdominal pleural muscles of the adult *Drosophila*. They show that the arborisations of the so-called pleural muscle motoneurons (PM-Mns), are accessible through metamorphosis. They establish a system whereby it is possible to genetically manipulate either synaptic partner and live-image the other. They start by defining the preparation and described it during metamorphosis. This is well done. (Not meaning to be self-referential (but only in part), but there are not unrelated studies by Consulas et al. from the Richard Levine Lab, Fernandes and VijayRaghavan on the development of flight muscle innervation and Sales and Hildebrand on the development of motoneurons in Manduca; All from prehistory, but some of them may be worth alluding to. Some deal with dendrites and some with neuromuscular arbors. Just a suggestion). The authors next study "The distribution of presynaptic components during axonal branch growth". This is done by using an RFP tagged Bruchpilot (BRP) driven by OK371-GAL4 (in the developing motoneurons). They next examine if synaptic activity is required for the stabilisations of arborisation seen and show that "PM-Mns elaborate their axonal arborisations without synaptic activity" This is tested by both blocking the vesicular glutamate transmitter, by showing that synaptic transmission does not take place before 60h APF while the neurons are activate-able by TRPA1 essentially after 60h APF. They conclude, after examining both pre- and post-synaptic components that "the changes in the organisation of synaptic machinery over the course of development suggest that the early accumulations of presynaptic components do not represent differentiated synapses." They next study the role of Neuroligin1-Neurexin in PM-Mn arborisation. They show, using mutants that there is an early role for Nlg1 signalling early in pleural neuromuscular development. Using elegant tools for generating and rescuing mosaics that show that the effects are caused by local interactions. Next, by examining the dynamics of expression of Ngl1::GFP and of NTX they show that the "Dynamic complexes of 'synaptic' adhesion proteins stabilise filopodia and drive branch growth. They similarly compared filopodia expressing Liprin-α::GFP and Strawberry tagged Syd1 and show a developing dynamic partnership during development. The authors next show that Nlg1-based adhesion complexes can direct a tropic mode of growth, they show this both by over-expression and ectopic expression. Finally the authors examine the "Physiological consequences of Nlg1 disruption".

This is a beautiful study, technically very well-executed, at the highest level, repeatedly seen from the work of the Williams lab. The videos are a delight to see. Aesthetics and technical quality apart, there is a very important conclusion. The authors show "for the first time outside of vertebrates" a 'synaptotropic-like' mode of arboriatons. While showing that synaptic activity has no role here they show that dynamic requirements of Neurexin, Neuroligin, Liprin-α and Syd-1, are necessary for stabilising filopodia during arborisation growth. They present support for generating a "spectrum of shapes and sizes by making only subtle changes to a 'stick and grow' algorithm. Their very tractable preparation now allows the determination of downstream mechanisms. Overall this is a robust and valuable study which advances the field significantly.

---

## [Author Response]

Reviewer #1:The manuscript by Constance and colleagues describes a novel imaging approach and molecular genetic analysis of the development branched axonal arbours of adult Drosophila motor neurons. The manuscript is well written, referenced and experiments clearly described. The manuscript makes two primary claims – that neuronal activity and vesicular neurotransmission have little or no influence on the development of early synaptic contacts in adult neurons and the Neuroligin-Neurexin signalling complex plays a key role. This first conclusion seems very well established by the experiments provided, fitting with an emerging theme of neurotransmission independence for most synapses in both invertebrates and vertebrates (excluding retinal bipolar neurons and a few other exceptions). The second conclusion is also reasonably supported and the authors have dealt well with a drawback of the system in that postsynaptic muscles are polynucleate limiting the effectiveness of clonal techniques. The majority of the manuscript makes use of a live imaging technique to evaluate synapse development. I would like to have seen this technique applied to Nlg and Nrx mutants more extensively to provide clearer insight into exactly which step of synaptogenesis is perturbed by these factors. Overall, even though the ultimate conclusions may not be entirely unexpected, the manuscript provides insight into a novel system to study synapse placement, assembly and development.Reviewer #1 (Minor Comments):The authors should carefully consider their use of the term "synaptotropic" to avoid confusion. The term has evolved to be used often to denote synapse growth and maturation rather than the original coinage.The number of animals and terminals examined should be added to some key observations e.g. Figure 6, Figure 7. as in the other legends.Reviewer #2:The authors establish a preparation to study in vivo the arborisation of neuronal processes. The preparation they use are the motoneurons that innervate the abdominal pleural muscles of the adult Drosophila. They show that the arborisations of the so-called pleural muscle motoneurons (PM-Mns), are accessible through metamorphosis. They establish a system whereby it is possible to genetically manipulate either synaptic partner and live-image the other. They start by defining the preparation and described it during metamorphosis. This is well done. (Not meaning to be self-referential (but only in part), but there are not unrelated studies by Consulas et al. from the Richard Levine Lab, Fernandes and VijayRaghavan on the development of flight muscle innervation and Sales and Hildebrand on the development of motoneurons in Manduca; All from prehistory, but some of them may be worth alluding to. Some deal with dendrites and some with neuromuscular arbors. Just a suggestion). The authors next study "The distribution of presynaptic components during axonal branch growth". This is done by using an RFP tagged Bruchpilot (BRP) driven by OK371-GAL4 (in the developing motoneurons). They next examine if synaptic activity is required for the stabilisations of arborisation seen and show that "PM-Mns elaborate their axonal arborisations without synaptic activity" This is tested by both blocking the vesicular glutamate transmitter, by showing that synaptic transmission does not take place before 60h APF while the neurons are activate-able by TRPA1 essentially after 60h APF. They conclude, after examining both pre- and post-synaptic components that "the changes in the organisation of synaptic machinery over the course of development suggest that the early accumulations of presynaptic components do not represent differentiated synapses." They next study the role of Neuroligin1-Neurexin in PM-Mn arborisation. They show, using mutants that there is an early role for Nlg1 signalling early in pleural neuromuscular development. Using elegant tools for generating and rescuing mosaics that show that the effects are caused by local interactions. Next, by examining the dynamics of expression of Ngl1::GFP and of NTX they show that the "Dynamic complexes of 'synaptic' adhesion proteins stabilise filopodia and drive branch growth. They similarly compared filopodia expressing Liprin-α::GFP and Strawberry tagged Syd1 and show a developing dynamic partnership during development. The authors next show that Nlg1-based adhesion complexes can direct a tropic mode of growth, they show this both by over-expression and ectopic expression. Finally the authors examine the "Physiological consequences of Nlg1 disruption".This is a beautiful study, technically very well-executed, at the highest level, repeatedly seen from the work of the Williams lab. The videos are a delight to see. Aesthetics and technical quality apart, there is a very important conclusion. The authors show "for the first time outside of vertebrates" a 'synaptotropic-like' mode of arboriatons. While showing that synaptic activity has no role here they show that dynamic requirements of Neurexin, Neuroligin, Liprin-α and Syd-1, are necessary for stabilising filopodia during arborisation growth. They present support for generating a "spectrum of shapes and sizes by making only subtle changes to a 'stick and grow' algorithm. Their very tractable preparation now allows the determination of downstream mechanisms. Overall this is a robust and valuable study which advances the field significantly.

We would like to thank the reviewers for highlighting the deficiencies in the manuscript and suggesting experiments. We think additional experimental work has helped clarify two key points in our model.

Firstly, we have mapped the changes in the distribution of Liprin-α and Bruchpilot during PM-Mn arbor development. These new data support our assertion that both proteins change their distribution relative to each other through time i.e. from being separate, with Liprin-α localized in puncta on the tips and along the length of filopodia, to being localized together within emerging terminal boutons.

Secondly, we have performed experiments looking at the localization of Liprin-α::GFP in a Neuroligin null background. We found that liprin still localises to growing branch terminals, including to the tips of some filopodia. On closer inspection, we find that Liprin-α::GFP puncta are largely absent from the unusually long and unbranched terminal arborisations found growing in the Nlg1 nulls. When we followed these branches they invariably collapse back or fail to grow any further.

In relation to changes in the text, we have made clear what we mean by the terminology ‘synaptotropic’. We have also cited the ‘earlier’ work on the development of the direct flight muscle NMJs and Rick Levine’s labs data.